# Nutrient-driven genome evolution revealed by comparative genomics of chrysomonad flagellates

Stephan Majda[1✉], Daniela Beisser[1] & Jens Boenigk[1]

Phototrophic eukaryotes have evolved mainly by the primary or secondary uptake of photosynthetic organisms. A return to heterotrophy occurred multiple times in various protistan groups such as Chrysophyceae, despite the expected advantage of autotrophy. It is assumed that the evolutionary shift to mixotrophy and further to heterotrophy is triggered by a differential importance of nutrient and carbon limitation. We sequenced the genomes of 16 chrysophyte strains and compared them in terms of size, function, and sequence characteristics in relation to photo-, mixo- and heterotrophic nutrition. All strains were sequenced with Illumina and partly with PacBio. Heterotrophic taxa have reduced genomes and a higher GC content of up to 59% as compared to phototrophic taxa. Heterotrophs have a large pan genome, but a small core genome, indicating a differential specialization of the distinct lineages. The pan genome of mixotrophs and heterotrophs taken together but not the pan genome of the mixotrophs alone covers the complete functionality of the phototrophic strains indicating a random reduction of genes. The observed ploidy ranges from di- to tetraploidy and was found to be independent of taxonomy or trophic mode. Our results substantiate an evolution driven by nutrient and carbon limitation.

[1] Department of Biodiversity, University of Duisburg-Essen, Essen, Germany. ✉email: stephan.majda@uni-due.de

Competition for nutrients and organic carbon are among the major ecological selective forces in the evolution of eukaryotes. The presence of phototrophic, heterotrophic, and mixotrophic taxa and evidence for multiple gains and losses of photosynthesis across nearly all major eukaryotic supergroups reflects the significance of nutritional constraints in the evolution of life[1–3]. Taxa that have experienced such an evolutionary modification of their basic nutritional strategy must bear hallmarks of this fundamental switch in their genomes. Here we track down the genomic fingerprints of the eco-evolutionary constraints linked to the varying significance of nutrient and carbon shortage.

In this paper, we refer to obligate heterotrophy and photo-trophy if not stated otherwise. Recent hypotheses suggest that the shift from photo- to mixotrophy was one way to overcome nutrient limitations while the shift towards heterotrophy was caused by carbon limitations[4,5]. In contrast to former individual case studies, here we use the parallel evolution of heterotrophic Chrysophyceae from phototrophic and mixotrophic ancestors in order to separate general directions and constraints in genome evolution from those related to the nutritional mode.

For phototrophic organisms, nutrients usually are a limiting factor. Phagotrophic uptake of bacteria and with that of nutrients and organic compounds could overcome limitations of essential nutrients such as nitrogen and phosphorus[6]. Alleviating the nutrient limitation may drive taxa into another challenge as nutrient shortage may simply be replaced by prey/carbon short-age. Phototrophs benefit from large cell sizes increasing the sur-face to maximize sun radiation, whereas heterotrophs profit from small cells, which enhance prey effectivity and enable feeding on ultramicrobacteria[4,5]. Genome sizes decrease from photo-trophic to heterotrophic chrysophytes[5]. This reduction minimizes the cost of nitrogen and phosphorus by saving nucleotides. Furthermore, the nucleotides thymine and cytosine differ by one nitrogen, making AT pairs cheaper than GC. Consequently, a low GC content could be caused by nutrient limitation. The changing relevance of either nutrient or carbon limitation alters constraints in genome evolution which should be reflected by the incor-poration of nucleotides with different costs, the loss of obsolete genes, and the evolution of new gene functions for a predatory lifestyle.

The Chrysophyceae within the Ochrophyta (Stramenopiles) are especially suited to investigate the evolutionary significance of this shift of the nutritional mode since the loss of photosynthesis occurred several times independently within this group[7–10]. Using transcriptome sequencing, Graupner et al.[11] analyzed plastid-targeting and -encoded genes of heterotrophic chryso-phytes compared to photo- and mixotrophs in which they identified different stages of plastid pathway reduction and degradation of accompanying structures. Dorrell et al.[10] extended the analysis by including plastid genomes and identified shared losses of function across chrysophytes. To our knowledge, all existing studies so far focused on the plastid and plastid-related functions but none yet on changes in the nuclear genome related to changes in trophic strategies. Findings from transcriptome sequencing of 18 chrysophyte strains provide first insights into molecular changes between trophic modes showing that hetero-trophs possess a reduced repertoire of genes related to photo-synthesis but an increased or upregulated repertoire of pathways associated with food uptake and motility[12]. Apart from this, changes in the nuclear gene content, genome size, GC content, ploidy, and further genomic features connected to trophy have not been analyzed. Therefore, in this study, we examine genomes of 16 chrysophytes including phototrophic, mixotrophic, and heterotrophic lineages to investigate the impact of the nutritional shift and its drivers.

We hypothesize that:

(1) The nuclear genome is reduced in size from photo- over mixo- to heterotrophic species, either as a result of nutrient limitations or as a size adaptation to feed on small bacteria. Likewise, the ploidy in heterotrophs is lower.

(2) Accompanying the genome reduction and specialization of heterotrophs, we will find a loss of functional genes, especially no longer required genes for photosynthesis, biosynthesis of certain amino acids, and plastid-targeting genes, as well as a decrease in intergenic regions, reflected by an increased gene density.

(3) The GC content of photo- and mixotrophic species is lower than in heterotrophic species if nutrients were the limiting factor during evolution.

## Results

**Genome assembly and gene content**. Sequencing generated at least 60 million 150 bp-long paired-end Illumina reads for each strain (see Table 1) and a total of 700,000 PacBio reads (details see Table S1). The comparison of binning strategies has shown that MetaBAT removed much more reads than MaxBin2 result-ing in one-third smaller assembly sizes (details see Table S2). Since the steps for filtering prokaryotic reads were identical resulting in similar qualities, we chose MaxBin2 for binning. On average 61.2% (median) of the sequencing reads were used for the assembly (see Table 1). Despite similar assembly sizes and N50 values between axenically and non-axenically cultured strains (see Table 1, 2), the BUSCO analysis revealed partly incomplete genomes. The completeness of recovered genes in the draft genomes was on average 59% (see Table 3). Assembly size and with that detection of genes and completeness of the genomes seems to be affected by the presence of bacteria; i.e., the quality of the assemblies was higher for axenic strains. However, the following analyses of ploidy, GC content, and gene density are independent of genome completeness.

We predicted 23,000 to 120,000 genes for the analyzed chrysophyte species (see Table 4). In assembled genomes, parts of long genes can possibly lie on several contigs and be counted multiple times thus overestimating the number of genes. We, therefore, clustered similar or duplicated genes to orthologous groups (OG) obtaining numbers in the range from 17,000 to 60,000, comparable to gene numbers obtained from transcrip-tome sequences of Chrysophyceae (8275 to 72,269[12], Ochromonas sp.: 19,692 genes[13]). Between 14 and 25% of these orthologous groups could be annotated with a KEGG Orthology (KO) ID for the assignment to the metabolic pathway. It is not surprising that additional use of PacBio sequencing and axenic strains lead to better assembly qualities (see supp. Table S3, Fig. S1). However, the omission of axenic and PacBio sequenced species leads to similar results concerning gene and pathway composition for the different nutritional modes. The pathway compositions regarding the KEGG functional groups differed mostly between single strains, instead of nutritional modes (see Fig. 1). Investigated pathways were almost complete with more genes present in the genome compared to the transcriptome (see Fig. 2, also Figs. S7 to S12 and Supplementary Data 1).

**Pan genome and core genome**. To expand the analysis of genes and pathways in single species, we compared the sets of ortho-logous groups (OGs) between the trophic modes. The pan gen-ome, the entire gene set of a group of species, as contrasted to the core genome, the intersection of genes within the group of species. The pan genome of all investigated phototrophic species contained 47,750 orthologous groups, while the pan genome of

**Table 1 Sequencing and assembly statistics.**

| Species | Strain | Reads [M] | GC [%] | Coverage | Used reads* [%] | N50 | Number of Contigs [10³] |
|---|---|---|---|---|---|---|---|
| C. sphaerica | JBC27 | 70 | 51.3 | 115.7 | 90.6 | 4039 | 35 |
| C. fuschlensis | A-R4-D6 | 77.9 | 51.7 | 38.1 | 22.8 | 2506 | 46 |
| P. encystans | 1006 | 83.7 | 54.5 | 13.9 | 6.2 | 1019 | 54 |
| P. encystans | JBMS11 | 59.7 | 51.4 | 52.2 | 11.9 | 3876 | 29 |
| P. lacustris | JBC07★ | 80.8★ | 53.1 | 116★ | 100 | 40,792★ | 9.1★ |
| P. lacustris | JBM10★ | 79.6★ | 53.1 | 128★ | 100 | 52,370★ | 9.4★ |
| P. lacustris | JBNZ41★ | 109.0★ | 52.9 | 153★ | 100 | 24,662★ | 13.8★ |
| S. vulgaris | 199hm | 79.8 | 47.9 | 19.9 | 14.9 | 2675 | 54 |
| C. nebulosa | CCAC 4401B | 66.5 | 43.9 | 45 | 50 | 1694 | 79 |
| D. divergens | FU18K-A | 64.3 | 45.1 | 60.1 | 70.7 | 1836 | 77 |
| D. pediforme | LO226K-S | 81.6 | 51.6 | 111.4 | 79.6 | 2530 | 50 |
| Epipyxis sp. | PR26K-G | 67.1 | 34.1 | 88.9 | 52.2 | 11,429 | 17 |
| C. danica | 933-7 | 88 | 45.4 | 298.6 | 100 | 97,456 | 14 |
| P. malhamensis | DS | 90.4 | 40.4 | 235.6 | 100 | 22,680 | 17 |
| M. annulata | WA18K-M | 67.3 | 40.1 | 57.5 | 61.8 | 7384 | 43 |
| S. sphagnicola | LO234KE | 112.5 | 46.9 | 88.9 | 61.2 | 2024 | 77 |

*The percentage of reads after binning and filtering out prokaryotic classified reads.
★Values from Majda et al.[18].

**Table 2 Genome size estimations.**

| Species | Strain | Assembly size (Haploid) [Mbp] | Estimated total size (Flow cytometry) [Mbp] | Ploidy |
|---|---|---|---|---|
| C. sphaerica | JBC27 | 82 | 157 | di |
| C. fuschlensis | A-R4-D6 | 70 | 143.2 | di |
| P. encystans | 1006 | 56 | 135 | - |
| P. encystans | JBMS11 | 61 | 182.2 | tetra |
| P. lacustris | JBC07★ | 49.4 | 155.8 | tri |
| P. lacustris | JBM10★ | 54.7 | 95.6 | di |
| P. lacustris | JBNZ41★ | 52.8 | 175.8 | tetra |
| S. vulgaris | 199hm | 90 | 293.6 | di |
| C. nebulosa | CCAC 4401B | 111 | - | tri |
| D. divergens | FU18K-A | 113 | 321.4 | di or tri |
| D. pediforme | LO226K-S | 88 | 226.6 | tri |
| Epipyxis sp. | PR26K-G | 59 | 192.6 | di or tetra |
| C. danica | 933-7 | 44 | 201.4 | tetra |
| P. malhamensis | DS | 58 | 150.8 | tri |
| M. annulata | WA18K-M | 109 | 671.8 | (di) |
| S. sphagnicola | LO234KE | 116 | 396.2 | di |

The genome size (in column 4) was estimated in ref. [5] by nuclear staining and flow cytometry.
★Values from Majda et al.[18].

**Table 3 BUSCO analysis results (protists set).**

| Species | Strain | Complete [%] | Singelton [%] | Duplicate [%] | Fragment [%] | Missing [%] |
|---|---|---|---|---|---|---|
| C. sphaerica | JBC27 | 21 | 8 | 13 | 3 | 76 |
| C. fuschlensis | A-R4-D6 | 57 | 36 | 21 | 16 | 27 |
| P. encystans | 1006 | 42 | 36 | 6 | 18 | 40 |
| P. encystans | JBMS11 | 62 | 42 | 20 | 12 | 26 |
| P. lacustris | JBM10 | 73 | 45 | 28 | 8 | 19 |
| P. lacustris | JBC07 | 69 | 36 | 33 | 9 | 22 |
| P. lacustris | JBNZ41 | 71 | 37 | 34 | 6 | 23 |
| S. vulgaris | 199hm | 61 | 58 | 3 | 4 | 35 |
| C. nebulosa | CCAC 4401B | 42 | 37 | 5 | 20 | 38 |
| D. divergens | FU18K-A | 74 | 70 | 4 | 12 | 14 |
| D. pediforme | LO226K-S | 31 | 19 | 12 | 23 | 46 |
| Epipyxis sp. | PR26K-G | 56 | 53 | 3 | 9 | 35 |
| C. danica | 933-7 | 96 | 86 | 10 | 0 | 4 |
| P. malhamensis | DS | 76 | 66 | 10 | 8 | 16 |
| M. annulata | WA18K-M | 68 | 65 | 3 | 12 | 20 |
| S. sphagnicola | LO234KE | 48 | 41 | 7 | 21 | 31 |

**Table 4 Gene prediction and orthologous groups.**

| Species | Strain | #OrthoGroups | #Genes | Annotated [%] |
|---|---|---|---|---|
| C. sphaerica | JBC27 | 30,998 | 62,845 | 23.1 |
| C. fuschlensis | A-R4-D6 | 33,640 | 78,807 | 23.1 |
| P. encystans | 1006 | 43,528 | 79,427 | 20.3 |
| P. encystans | JBMS11 | 30,479 | 54,910 | 19.6 |
| P. lacustris | JBM10 | 34,008 | 42,266 | 24.8 |
| P. lacustris | JBC07 | 35,720 | 41,883 | 24.7 |
| P. lacustris | JBNZ41 | 38,877 | 45,773 | 24.6 |
| S. vulgaris | 199hm | 35,008 | 77,543 | 21.2 |
| C. nebulosa | CCAC 4401B | 55,495 | 91,636 | 15.8 |
| D. divergens | FU18K-A | 62,902 | 93,394 | 13.7 |
| D. pediforme | LO226K-S | 39,124 | 80,221 | 23.2 |
| Epipyxis sp. | PR26K-G | 17,019 | 23,363 | 23.5 |
| C. danica | 933-7 | 23,539 | 36,312 | 22.5 |
| P. malhamensis | DS | 21,158 | 33,769 | 21.8 |
| M. annulata | WA18K-M | 35,854 | 65,203 | 17.5 |
| S. sphagnicola | LO234KE | 34,689 | 119,235 | 21.8 |

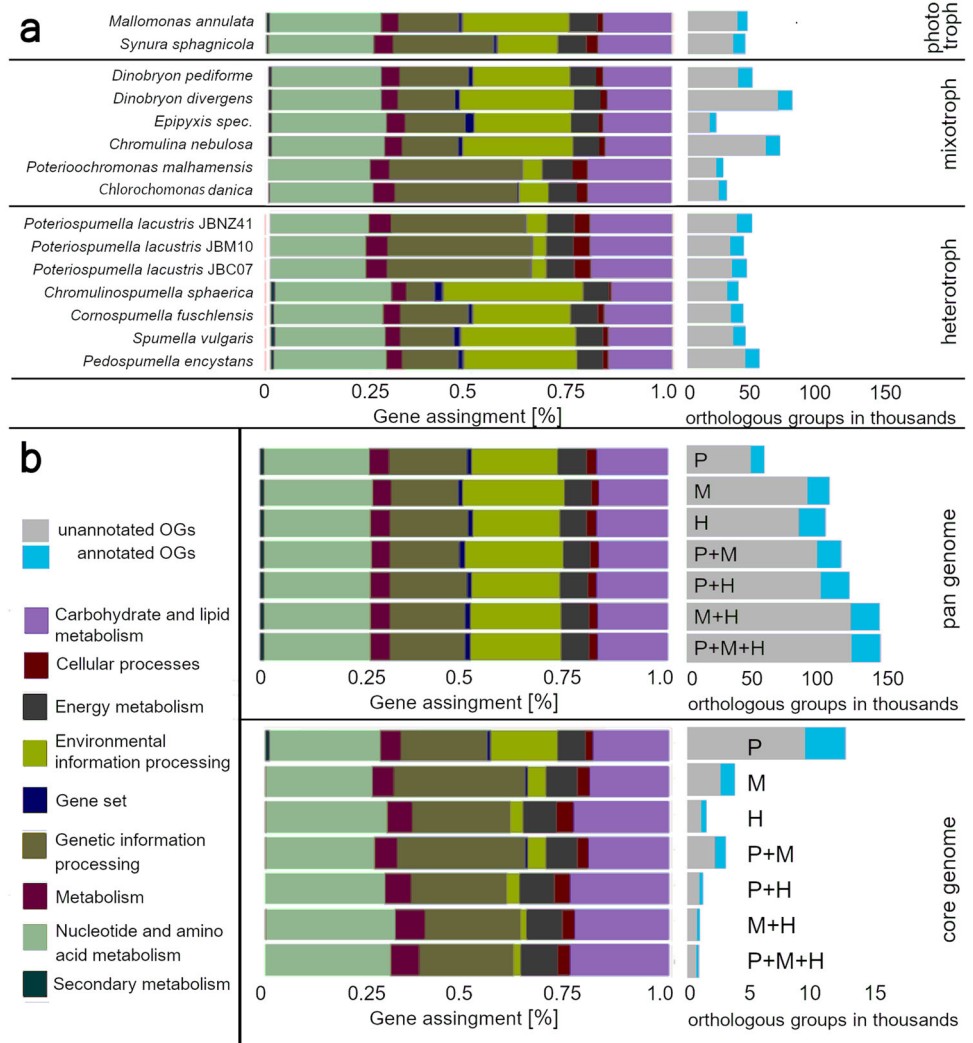

**Fig. 1 Scope and gene composition of specific groups.** Genes were clustered to orthologous groups (OGs), whose composition is based on gene assignment with the KEGG top hierarchy level. The total amount of OGs is shown by gray (unannotated) and light blue (annotated) bars. **a** Single strains. **b** Combination of phototrophic (P), mixotrophic (M), and heterotrophic (H) strains. The pan genome contains every gene of the specific combination, whereas the core genome consists only of genes present in each strain of the group. See supp. Table S6 for values.

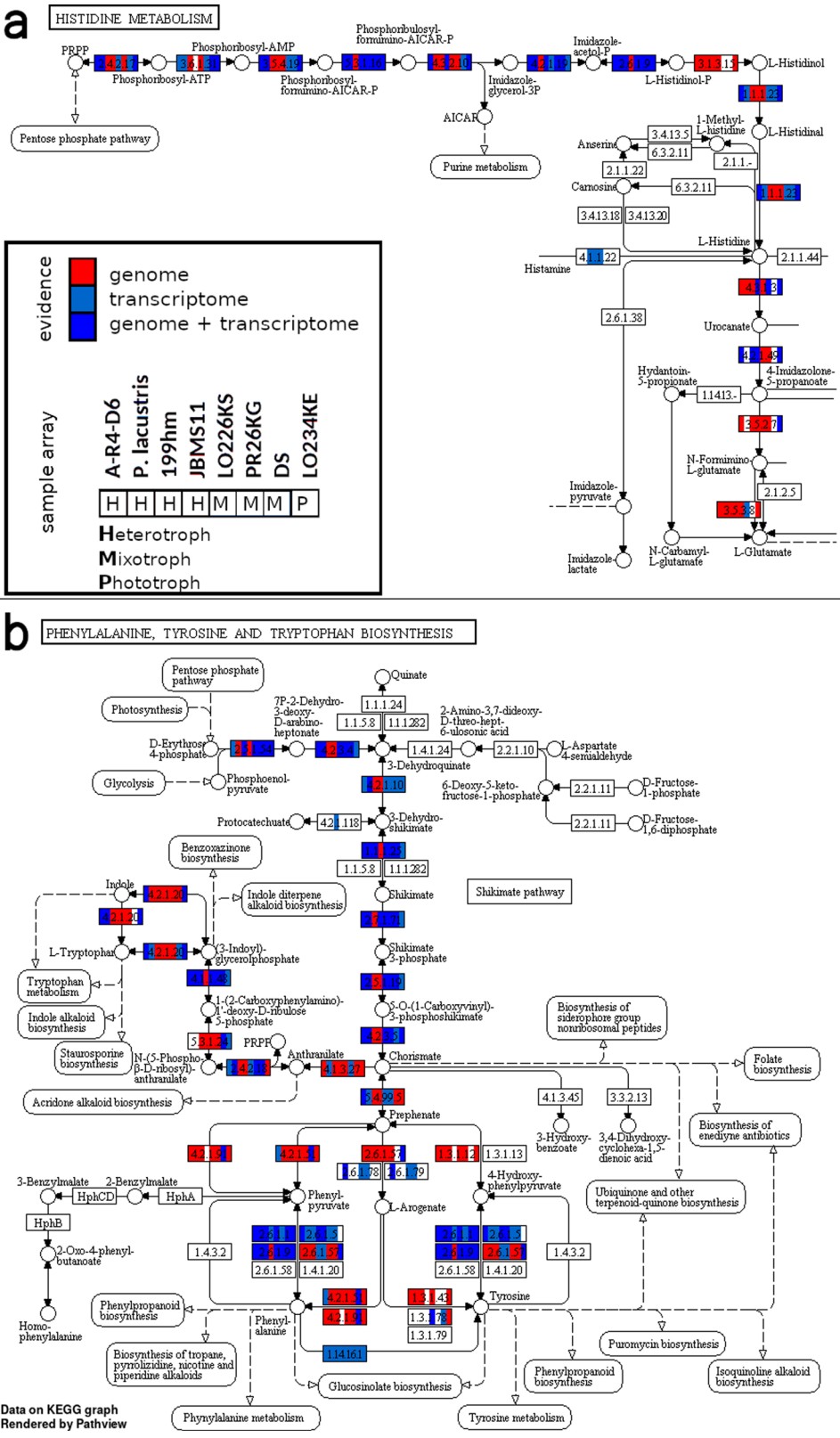

**Fig. 2 Pathway completeness.** Each box represents a gene and is divided into eight segments. The segments represent from left to right the strains: A-R4-D6, *P. lacustris* (pool of JBC07, JBM10 and JBNZ41), 199hm, JBMS11 (heterotroph, red-marked in the legend), LO226KS, PR26KG, DS (mixotroph, blue marked), and LO234KE (phototroph, green marked). The segment is colored based on the evidence (red: genome, bright blue: transcriptome, dark blue: both). **a** The path from PRPP (phosphoribosyl pyrophosphate) to histidine was incomplete for 199hm and JBMS11 based on transcriptomic data, but could be completed by genomic data. Whereas from histidine to glutamate was neither evidence for strain DS. **b** Genomic data complete pathway between 3-Dehydroquinate and Chorismate for strains 199hm and JBMS11.

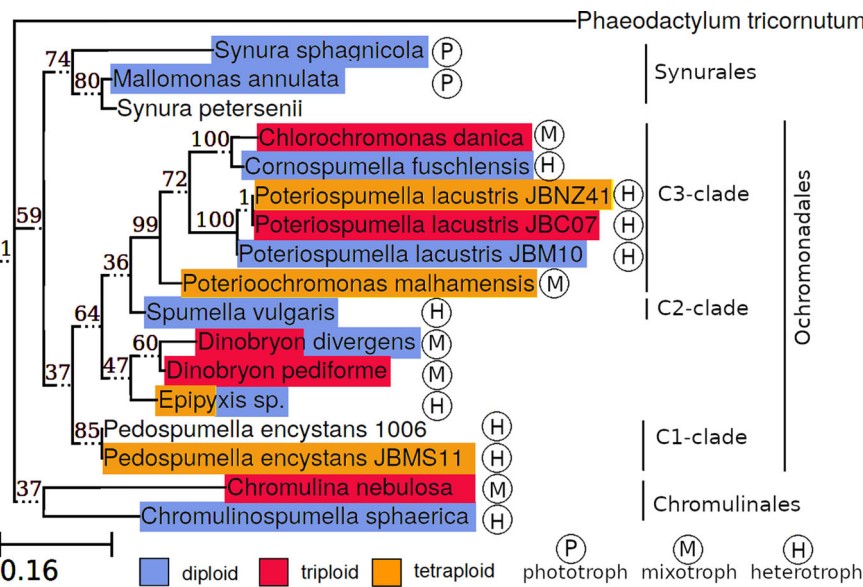

**Fig. 3 Taxonomic distribution of ploidy.** The ploidy (blue = diploid, red = triploid, and yellow = tetraploid) is independent of nutritional mode (phototroph, mixotroph, and heterotroph) or taxonomy. A phylogenetic tree is based on 18S rRNA gene sequences.

mixotrophic taxa contained 90,463 OGs and that of heterotrophic taxa 83,503 OGs. Combining the pan genomes of two trophic modes increased the total number of OGs (Fig. 1), while the combination of all trophic modes resulted in a similar number of OGs as the combination of mixotrophs and heterotrophs only. Each combination of orthologous groups of different nutritional modes resulted in similar compositions of KEGG functional groups at the top hierarchy level (see Fig. 1) and the next lower level (see Fig. S3). In contrast to the pan genome, the intersection between the OGs of phototrophic species resulted in the largest core genome (phototroph OGs: 8,934; mixotroph OGs: 2561; heterotroph OGs: 1084), as a consequence of the small group size, while the core genome of the heterotrophic species was the smallest.

**Genome size**. Considering only genomes with high completeness, hetero- and mixotrophic species did not differ largely in assembly size, ranging at around 67 Mbp. We observed significant differences in assembly size between all three groups (ANOVA $p$-value < 0.05), but these differences were mainly due to the substantially larger genomes of the phototrophic species (see Table 2). Due to the few species in this group, we refrain from drawing a definite conclusion here.

**Ploidy**. The ploidy of the investigated species ranged from diploidy to tetraploidy and seemed to be independent of nutritional mode and taxonomy (see Fig. 3). *Chromulinospumella sphaerica*, *Cornospumella fuschlensis*, *Poteriospumella lacustris* strain JBM10, *Spumella vulgaris* and *Synura sphagnicola* were diploid, *Poteriospumella lacustris* strain JBC07, *Chromulina nebulosa*, *Dinobryon pediforme,* and *Poterioochromonas malhamensis* were triploid and *Pedospumella encystans* strain JBMS11, *Poteriospumella lacustris* strain JBNZ41 and *Chlorochromonas danica* were tetraploid (see Table 2) For *Mallomonas annulata* we had only poor indications of diploidy and the ploidy level were ambiguous in *Dinobryon divergens* (di- or tetraploid) and *Epipyxis sp.* (di- or triploid) (see Fig. S5). For *Pedospumella encystans* 1006 data was not sufficient to obtain a clear result.

**Gene density**. In general, there is a correlation between genome size and gene density in eukaryotes[14]. We expected to find a correlation between nutritional mode and gene density. Especially, heterotrophs should have a high gene density, because of the strong selection towards smaller cells. The average gene density ranged from 1031 (phototroph), over 1049 (mixotroph) to 1520 (heterotroph) genes/Mb. A slight trend towards higher gene density with advanced heterotrophy was visible, but these differences were not significant ($p$-value = 0.32, see Fig. 4).

**GC content**. The mean GC content of the phototrophic (43.5%) and mixotrophic group (43.4%) was similar, whereas heterotrophs (51.6%) had a strongly increased GC content (ANOVA: $p$-value = 0.0041, posthoc test mixo- versus heterotrophs $p$-value = 0.0053, see Fig. 4).

We separated the total GC content based on coding sequences, introns, and non-coding sequences (see Table S4). The mean GC content of noncoding regions was smallest (35.5%), intron GC content was in between (41.4%) and coding sequences had the highest GC content (54.8%) (pairwise t-tests, each $p$-value ≤ 0.001). Using only the third position of the coding sequences (GC3) resulted in a GC value of 52.6%. The coding, GC3, intron, and non-coding GC content were independent of the nutritional mode (ANOVA: $p$-value > 0.05). Significant dependencies were also not apparent between assembly size, or total genome size, and the GC content ($p$-values > 0.5).

## Discussion

The comparative genome analysis of chrysophytes revealed genome reduction accompanied by an increase in the genomic GC content and an increase in gene density as a concomitant phenomenon of the evolutionary switch from phototrophy to heterotrophy. Gene reduction occurred in all heterotrophic lineages but different genes were lost in the various phylogenetic lineages presumably reflecting differential evolutionary selective pressures and adaptations to different environments in the distinct lineages. Even though genome reductions evolved independently, gene losses were predominately observed for non-annotated genes; i.e., genes of unknown functions, while genes and pathways of the primary metabolism were kept in all lineages.

We found considerable differences in genome size and ploidy between strains. While genome size reduction seemed to be related to the change of trophy but independent of phylogeny, a

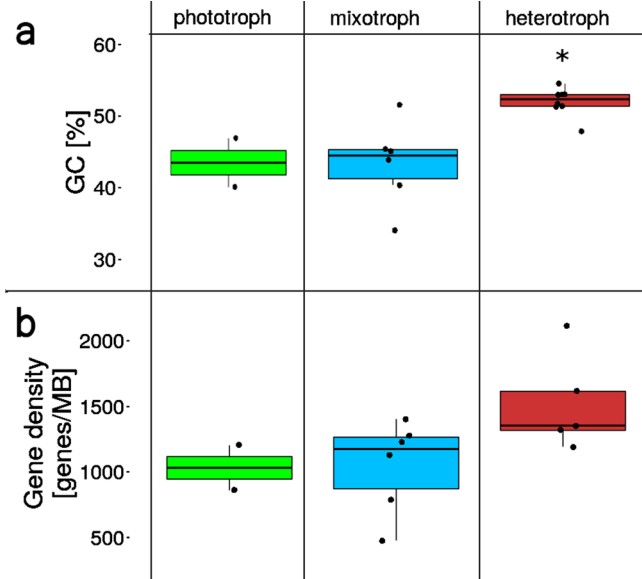

**Fig. 4 GC content and gene density in relation to nutritional mode. a** The GC content depends on the nutritional mode ($p < 0.001$). The GC content of the heterotrophic group is significantly different from the mixotrophic and phototrophic groups (marked with an asterisk). Error bars represent standard deviations. **b** There is no significant correlation between trophic mode and gene density. Phototroph: green, $n = 2$; mixotroph: blue, $n = 6$; heterotroph: red, $n = 8$ biologically independent samples.

variation of ploidy was found in different lineages and seemingly independent of the nutritional mode. From the assembly sizes, we could not definitively assess differences attributable to the different trophic modes, but the two larger phototrophic genomes suggest it (see Table 2). Nucleic acids are amongst the most phosphorus and nitrogen-demanding biomolecules and large genomes are costly to build and maintain, thus under nutrient limitation a reduction in genome size is expected. Further, the genome size has been shown to correlate with cell volume (e.g., ref. [15]), which is reduced in phagotrophic species preying on ultramicrobacteria. This is supported by a study from Olefeld et al.[5] analyzing 46 chrysophyte strains, in which a reduction in cell volume and genome size from photo- over mixo- to heterotrophs could be clearly shown with flow cytometry. In this study[5], the size of the phototrophic strains was considerably larger, although it should be noted that this applies to algae with secondary plastids, while species with primary plastid may be much smaller e.g., *Ostreococcus tauri*[16].

The study from Olefeld et al.[5] calculated haploid genome size based on the assumption of diploid genomes in chrysophytes. As we show here, ploidy varies between strains and thus variation in ploidy overlays the general trend of genome size reduction. Taking the different ploidies into account the genome size estimates based on flow cytometry[5] and based on assembled genomes (this study) largely correspond except for few strains for which the ploidy level could not be unambiguously determined (Table 2). In particular, the genomes of phototrophic strains were considerably larger as compared to those of heterotrophic and mixotrophic strains. Since the two methods do not match consistently, we rather rely on the genome size results from flow cytometry. We compared gene density and show statistical results for both methods in the supplement (see supp. Table S5, Fig. S13). In no case, we could find a significant difference between the three trophic groups concerning gene density.

Ploidy varied between diploidy and tetraploidy and was independent of phylogeny and nutritional mode. The independence

from phylogeny was not surprising as different ploidy levels have already been demonstrated within one species[17,18]. Apparently, a change in ploidy occurs frequently and is often associated with broader ecological niches and/or invasiveness due to increased heterozygosity and flexibility[19]. In particular, the extra gene copies in polyploid genomes may be beneficial for evolving new functions, novel gene combinations, and modified gene expression. For instance, flowering plant lineages polyploidize at a rate that is about 2–10% of the speciation rate[20,21]. In protists, which replicate predominantly asexual, the rate could even be higher, since the disadvantages of vulnerability to infertility or pairing difficulties in meiosis[22] are of little relevance. Furthermore, polyploidy is advantageous with respect to gene redundancy[22], especially in predominantly asexual species in order to prevent the accumulation of mutations (Muller's ratchet)[23]. However, not only increases in ploidy but also a decrease of the ploidy level may frequently occur as it has been demonstrated for tetraploid or triploid yeast within a short time period of only 186 generations[24]. Irrespective of the potential advantages of polyploid genomes, an increase in ploidy level counteracts the attempt of reducing genome size since cell size and DNA content usually correlate[25,26]. Polyploid organisms form larger cells[27,28], but the reason for this correlation is not yet clearly understood. Both, the higher amount of DNA and the higher expression of proteins have been suggested to be causative[29,30], however, deviations thereof are known even within the same genus[31]. These deviations imply that more factors are decisive for the cell size. Likewise, even though plants often increase their cell size by increasing their ploidy level[28], preventing the ploidy enhancement did not affect cell size[32]. Summarizing, our data point to a trade-off between the advantages of genome size reduction and polyploidization: Cell size reduction—and with that genome size reduction—is discussed as an adaptation to increase the efficiency in preying on small bacteria, in particular on ultramicrobacteria. At the same time, genome size reduction lowers the costs of DNA buildup and reproduction. In contrast, polyploidization may allow for higher flexibility enabling the taxa to populate broader niches and may prevent or at least slow down the accumulation of mutations. When nutrients are not limiting this trade-off should level off as it has also been shown for cultures with repeated transfer[17], since selection pressure decrease under laboratory conditions. However, due to the selective advantages of both, genome size reduction and polyploidy, it seems reasonable that polyploidy may also evolve under nutrient limitations despite the extra cost for genome maintenance.

To analyze the gene composition and gene loss we compiled for each nutritional group all combinations of the pan genome and the core genome. The pan genome comprises genes present in at least one representative, while the core genome comprises genes present in all representatives. One major finding of our study is the deviation between the core and pan genome of the heterotrophic taxa. The small core genome in contrast to the large pan genome of the heterotrophs implies that the strong genome reduction in the evolution of the heterotrophs was accompanied by a strong niche specialization. Even though numerous genes have been lost in each heterotrophic lineage the distinct lineages lost different genes. Phylogenomic patterns were consequently only found in closely related lineages, i.e., loss of similar genes and pathways (in particular in the clade comprising *Poteriospumella*, *Poterioochromonas*, and *Chlorochromonas*). Further, our data demonstrate that in all trophic groups some of the genes are not found in at least one of the other trophic groups. However, combining the group of mixotrophic and heterotrophic species, their genes cover almost all genes and pathways found in the phototrophic species. We expected to see a gradual reduction of pathways related to photoautotrophy

depending on the stage of nutritional mode. This gene loss pattern has been shown for the plastome, even if exceptions such as genes related to Rubisco could still persist[33]. The study of Graupner et al.[11] suggests that for pathways directly linked to photosynthesis and genes encoded in the plastid a certain sequence of gene losses occurs, accompanying the shift from mixotrophy to heterotrophy. Here we show that this is not the case for the majority of nucleus-encoded genes but gene loss seems to be different for distinct lineages indicating that different strains presumably were exposed to different evolutionary forces. This is supported by an increased fraction of genes related to environmental information processing and secondary metabolism in the photo- and mixotrophic group (see Fig. 1) while the heterotrophic species have a more specialized and unique gene inventory in these functional groups.

We further demonstrate that primary pathways and annotated pathways of the secondary metabolism are almost complete in the genome. Gene reductions concern mostly genes that could not be annotated; i.e., genes of unknown function which presumably are important for niche specialization but not for the basic requirements of the cells. For instance, the pathways for the biosynthesis of amino acids were complete in all investigated strains even though amino acids can, in principle, also be taken up with food in the phagotrophic taxa (see Fig. 2, also Figs. S7 to S12). This completeness of pathways seems to be different for plastid-encoded genes as suggested by transcriptome data and supported by the structural reduction as observed in microscopical analyses[9]. The maintenance of the pathways of the primary (and to a large part also of the secondary) metabolism could be advantageous for backing-up nutritional requirements in the case of food shortage. As we analyzed the occurrence of genes, our results do not necessarily imply that all genes are functional. But the fact that pathways were mostly found to be complete supports this assumption. Still, undetected mutations could have rendered genes non-functional or modified their function, for certainty, additional verifications would be needed.

There is a small trend towards higher gene density with advanced heterotrophy, but possibly due to the small sample size, the difference between groups was not significant. The gene densities were generally high compared to gene densities of other organisms. Stramenopiles have on average a gene density of 200–400 genes/Mb[18], whereas prokaryotes or archaea typically have around 1000 genes/Mb. As eukaryotes with small genomes are known to have gene densities similar to bacteria[14] and constant gene densities independent of genome size[34] our estimates still seem reliable. However, this high gene density could partly be due to an overestimation in the gene prediction, in particular, due to the merging of repeat regions during assembly and the potential presence of genes on both strands. Since the gene density between strains sequenced only with Illumina is comparable to strains additionally sequenced with Pacbio, a bias due to collapsed repeat regions should be very small.

The GC content is known to be correlated with genome size in bacteria but the relationship for other kingdoms is less clear. Studies analyzing genome size and base composition in available sequenced genomes in various kingdoms found a correlation between average GC content and genome size which was positive in bacteria (specifically Proteobacteria and Actinobacteria), weakly positive in Ascomycota fungi and some plants, but negative in animals and indifferent in two analyzed protistan phyla[35–37]. We show that the GC content of chrysophytes did not correlate with genome size. In contrast, the total GC content was significantly different between nutritional modes. The total GC content differs due to several factors, which were not significant individually (e.g., gene density, the difference between nutritional modes in GC content in introns, coding

sequences, and non-coding sequences), but contributed jointly to the measured difference. While the mean GC content of the phototrophic and mixotrophic groups was similar, the heterotrophs showed a strongly increased GC content. In general, an increased GC content is associated with higher needs regarding nutrients and energy[38,39]. Adenine and guanine require 5, cytosine 3, and thymine/uracil only 2 nitrogen atoms, thus the difference in costs is observable in the GC content of a genome. Species living in or adapted to nitrogen-limited environments therefore often use nucleotides that require fewer nitrogen such as A and T[40]. In addition, at least in bacteria, organisms show distinct GC patterns that are not explained by phylogeny but by similar environments[41]. It has been shown that plants that require more nitrogen to conduct photosynthesis experience stronger selection to minimize nitrogen biosynthesis costs[42]. As the evolution towards heterotrophy; i.e., the establishment of an alternative nutrient source, is typically related to adaptations to low nutrient availability, a lower GC content should be expected in phototrophic chrysophytes. The evolutionary constraint of nutrient limitation should be already relaxed in mixotrophic species, but as the photosynthetic apparatus is costly and accounts for about half of the cells proteins[43], mixotrophs may be subject to similar constraints as the phototrophs. Heterotrophic species should predominantly be limited by carbon, not by nutrients, and thus can afford a higher GC content. The increased energy demand of high GC content led to a low GC content in non-coding regions and introns, whereas coding sequences remained GC-rich encoding cheaper amino acids[38].

## Conclusions

We demonstrate that nutritional constraints drive genome evolution and that limitations in nutrient and carbon acquisition leave footprints in genome evolution. Based on comparative genome analysis of lineages that independently and in parallel evolved heterotrophy from mixotrophic ancestors we separated genomic footprints linked to a nutritional switch from random shifts. In particular, genome size reduction and a shift in GC content reflect changes in nutrient limitation during the evolution of obligate heterotrophy from phototrophic and mixotrophic ancestors. Gene losses accompany these changes but vary between lineages indicating a presumably increasing niche separation and differentiation between the distinct heterotrophic lineages. In the broader context of eukaryotic mega evolution, the discovery of disparate gene losses in different lineages and its implication of an accelerated differentiation is intriguing as it adds a new facette on the evolution of eukaryotic diversity. Nutrient and carbon limitation may not only be crucial for the transformation of feeding strategies but furthermore speed-up the evolutionary diversification and thus contribute to rapid radiations on short evolutionary time scales.

## Methods

**Cultivation and sequencing**. We cultivated and sequenced 16 strains (see Table 5) according to Hahn et al.[44] and Majda et al.[18]. Non-axenic heterotrophic and mixotrophic cultures were grown with bacterial food supply (*Limnohabitans planktonicus*; strain IID5T). Two days before DNA harvesting in these cultures the feeding with bacteria was omitted.

**Genome assembly and binning**. Unless otherwise stated, the default settings were used for the following programs. An automated workflow with Snakemake[45] processed the sequencing data. Supplementary Fig. S2 gives an overview of the assembly and binning procedure. Reads were quality checked, filtered and adapters were removed by the sequencing provider, further, the read quality was checked by FastQC (v0.11.5; with a Phred quality score criteria of around 10 for PacBio reads and above 20 for Illumina reads[46]). SPAdes (v3.13.0; with parameters: –meta[47]) assembled the Illumina reads. If PacBio reads were available they were incorporated in the SPAdes assembly and additionally assembled with CANU (v1.8; with

**Table 5 General information of examined species.**

| Species name | Strain | Nutritional mode | Reference | Sequenced[†] | Medium |
|---|---|---|---|---|---|
| *Chromulinospumella sphaerica* | JBC27 | heterotroph | 9 | I | IB |
| *Cornospumella fuschlensis* | A-R4-D6 | heterotroph | 9 | I | IB |
| *Pedospumella encystans* | 1006 | heterotroph | 68 | I | IB |
| *Pedospumella encystans* | JBMS11 | heterotroph | 68 | I / P | IB |
| *Poteriospumella lacustris*★ | JBC07 | heterotroph | 68 | I | NSY |
| *Poteriospumella lacustris*★ | JBM10 | heterotroph | 68 | I / P | NSY |
| *Poteriospumella lacustris*★ | JBNZ41 | heterotroph | 68 | I | NSY |
| *Spumella vulgaris* | 199hm | heterotroph | 69 | I | IB |
| *Chromulina nebulosa* | CCAC 4401B | mixotroph | 69 | I | WC |
| *Dinobryon divergens* | FU18K-A | mixotroph | 70 | I | WC |
| *Dinobryon pediforme* | LO226K-S | mixotroph | 71 | I | IB |
| *Epipyxis sp.* | PR26K-G | mixotroph | 72 | I | WC |
| *Chlorochromonas danica*★ | 933-7 | mixotroph | 73,74 | I / P | NSY |
| *Poterioochromonas malhamensis*★ | DS | mixotroph | 75 | I / P | NSY |
| *Mallomonas annulata* | WA18K-M | phototroph | 76 | I | WC |
| *Synura sphagnicola* | LO234KE | phototroph | 77 | I / P | WC |

[†]I = Illumina Hiseq XTen, P = PacBio RSII.
★Axenic cultures, processed according to Majda et al.[18].

parameters: genomeSize = 100 m, correctedErrorRate = 0.105[48]), whereby the genome size estimations from Olefeld et al.[5] were used if possible. We renounced from genome size estimation based on *k*-mers, since it often led to size bias in chrysophytes[18], especially through read filtering in non-axenic strains. To classify the reads we combined a tool using compositional features and other applying taxonomical methods. The tool MaxBin2 (v2.2.5[49]) binned the contigs created by SPAdes. Subsequently, Kraken2 (v2.0.7[50]) classified the bins taxonomically. Therefore, the NCBI database (Release 2018-11-01) and the assemblies of the axenic cultures *Chlorochromonas danica*, *Poteriooichromonas malhamensis*, and *Poteriospumella lacustris* were used as a reference. Including the axenic cultures enabled to classify chloroplast bins as eukaryotic instead of prokaryotic and prevented their exclusion. Bins consist out of several contigs. In some cases, the bins contain contigs classified as eukaryotic, as well as prokaryotic or unclassified. Ambiguous cases are very rare (less than 5%), but we have nevertheless defined a criterion for this. Depending on the bin composition, we classified and processed unclassified contigs as follows:

1. If the number of eukaryotes was at least half the number of bacteria, the unclassified were defined as eukaryotic.
2. If the number of unclassified contigs was twice as large as the number of bacteria, the unclassified were defined as eukaryotic.

We have defined these points because the number of known bacteria in the NCBI database is much higher than that of related protists.

In addition, the binning tool MetaBAT(v2.12.1[51]) was tested followed by the same classification and filtering steps. The bins were merged into two files containing either eukaryotic or bacterial sequences. The Illumina reads were aligned with Bowtie2 (v.2.3.0[52]) against the bacterial contigs. Subsequently, the hits were mapped again with Bowtie2 against the eukaryotic contigs. Unaligned reads were marked as bacterial and excluded from further processing. Normally, reads mapping to both the bacterial and the eukaryotic contigs were randomly assigned. To keep the eukaryotic reads in this case we used incremental mapping.

The contigs of the CANU assembly were only binned by Kraken2, as the long contig length caused sufficient classification accuracy. In a second step, we filtered out all contigs that were not eukaryotic. We repeated the SPAdes assembly (without –*meta* parameter) with filtered Illumina reads and filtered PacBio reads if available (parameter: –untrusted-contigs). Contigs smaller than 500 bp were discarded.

Finally, the Benchmarking Universal Single-Copy Orthologs (BUSCO) software (v4.0.6 with parameters: –lineage_dataset stramenopiles_odb10 –mode prot[53]) was used to verify the existence of essential genes.

**Ploidy estimation**. The genome *k*-mer frequencies were counted by KMC tools (v3.1.0 with parameters: -k 21[54]). Genome ploidy was estimated based on these *k*-mer frequencies with *smudgeplot* (v0.1.3 with parameters: -k21 -m300 -ci1 -cs10000[55]). Thereby, the ratio between *k*-mer pairs, which differ by one nucleotide, was determined. This ratio is characteristic for a specific ploidy (1/2 = diploid, 1/3 = triploid, etc.; see supp. Figs. S4 to 6).

**Gene prediction**. The tool AUGUSTUS (v3.3; with parameters: –gff3=on –progress=true –singlestrand=true –UTR=off –species=arabidopsis[56]) was used

for gene prediction. For the strains *199hm, A-R4-D6, JBMS11, LO226K-S, LO234KE, PR26KG* and *DS* transcriptomic data ([12], European Nucleotide Archive (ENA) Project ID: PRJEB13662) supported the gene prediction. In this case, RepeatScout (v1.0.5[57]) was used to extract repetitive sequences for each strain. Subsequently, the genomes were masked by BEDTools (v2.28, with parameters: maskfasta -soft[58]). This prevented aligning RNA reads or predicting genes in repeat regions. Finally, AUGUSTUS parameter were modified to: –softmasking=1 –species=arabidopsis -gff3=on -singlestrand=true -UTR=off –alternatives-from-evidence=false –extrinsicCfgFile –hintsfile. In the extrinsicCfgFile the following relevant weighting was changed to weight more strongly the supporting transcriptomic sequences:

exon hit bonus: 1e + 10 (instead of 1), nonexonpart malus: 2 (instead of 1) Aligning the transcriptome data with the genome by Minimap2 (v.2.9-r720; with parameters: -c -L -x splice -G 80K -t 16[59]) created the hintsfile.

**Gene annotation and clustering**. Genes were annotated according to ref. [18] by aligning the predicted genes to the KEGG (Release 2014-06-23,[60]) and UniProt database (Release 2017-09-18[61]) with Diamond (v0.9.10.111[62]).

OrthoFinder (v2.2.6; with parameters: -S diamond) clustered the protein sequences of all strains to *orthologous groups* (OG). The majority of gene annotations within an orthologous group determined the annotation of the whole group. We built the union (all OGs) and intersection (OGs shared among all) between species of one nutritional mode (phototroph, mixotroph, and heterotroph) and between nutritional modes. In addition, for these sets, the composition of KEGG functional groups was determined.

**GC content and gene density**. The GC content was assessed for the whole genome and for intergenic regions. Significant differences in gene density and GC content between the nutritional modes were calculated using analysis of variance (ANOVA) in R. The gene density ($d$) was calculated for the number of genes ($n$) by the formula:

$$d = \frac{n}{\sum bp_{contigs > 500bp}} \cdot 10^6 \tag{1}$$

**Pathway analysis**. Present annotated genes were visualized in the KEGG pathway maps by the R package Pathview (v1.24.0[63]). We focused on the following pathways: biotin metabolism, histidin metabolism, tryptophan metabolism, nitrogen metabolism, lysine biosynthesis, pyhenylalanine, tyrosine, and tryptohan biosynthesis, thiamine metabolism and valine, leucine, and isoleucine biosynthesis.

In addition, a comparison was made to determine whether genes from a transcriptome dataset[12] were also present.

**Phylogenetic tree**. The 18S sequence of each strain, *Phaeodactylum tricornutum* (strain: KMMCC B-386, accession number: GQ452860.1, as outgroup) and *Synura petersenii* (strain: CCMP857, accession number: EF165117.1) were aligned with the Clustal Omega algorithm[64] and processed with EMBLs Simple Pyhlogeny tool[65]. We calculated maximum likelihood with W-IQ-TREE (v1.6.11; with parameters: -st DNA -bb 1300[66]) and visualized the tree with ETE[67].

**Reporting summary**. Further information on research design is available in the Nature Research Reporting Summary linked to this article.

## Data availability

Whole-genome sequencing data that support the findings of this study are available from NCBI under the accession numbers: PRJNA546545 and PRJNA548251; PRJNA546545 and PRJNA548251.

## Code availability

https://github.com/stephanmajda/genome_analysis_pipeline https://github.com/stephanmajda/Genome_Transcriptome_to_GFF.

Each repository contains a file containing software and version requirements.

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

## Acknowledgements

We thank Micah Dunthorn for proof-reading and reviewing. We acknowledge support by the Open Access Publication Fund of the University of Duisburg-Essen.

## Author contributions

J.B. conceived the study; J.B., S.M. designed the lab experiments; S.M. and D.B. designed the computational procedure; S.M. performed the experiment and analyzed the data.; S.M., J.B., and D.B. interpreted the data; S.M., J.B., and D.B. wrote the manuscript; all authors read and approved the final manuscript.

## Funding

## Competing interests

The authors declare no competing interests.
