## [Peer Review File · Communications Biology]

Reviewers' comments:

Reviewer #1 (Remarks to the Author):

The manuscript reports on a genomic comparative study of 16 chrysophyte strains. The strains without functional chloroplasts were obligate heterotrophs and ones with chloroplasts were considered either mixotrophic or phototrophic. Mixotrophy and heterotrophy were defined as being phagotrophic (able to capture and use bacteria as nutrient or carbon sources). The main conclusion was that the chrysophytes moved towards heterotrophy by independent loss of suites of genes. The manuscript addresses an interesting question. The main weakness of the study is the lack of completeness (coverage) of the genomes, which were estimated to be from 3.7% to 56.3% complete using BUSCO. A second problem is relatively poor taxon sampling, with a heavy bias towards heterotrophic strains. The phototrophic strains both belong to more distantly related taxa and it seems that phylogeny might better explain the differences between the phototrophic and mixo-hetero species. This shortcoming could be difficult to address since the ancestral state of chrysophytes may well have been mixotrophy, and purely photosynthetic species lost the capacity to engulf bacteria.

The sequenced phototrophic species were both scaly flagellates (Silica scales) and one could hypothesize that the use of silica covering results in loss of phagotrophy in general. The main evolutionary driver could be anti-predation (or something else) and nothing to do with nutrient limitation.

The identification of pathways, which eluded earlier transcriptomic results, is interesting given the recent reports relying on transcriptomes generated under laboratory growth conditions. The pathways are the strong point of the manuscript. But the use of dark blue background and black letters in the graphic made it difficult to examine the results in detail.

Specific comments:

Introduction.

P3. Line 12. Even if a point modification was "random", for the resulting phenotype or trait to be maintained, there would have to be some evolutionary selection.

Line 15-19. Some linking (context) text seems to be missing. Explain how nutrient limitation could be related to "size"?

Line 22. This would be a good place to explain what is meant by "costs" I assume this is relative content of N or P in nucleic acids or amino acids (codon usage). Since the sequenced chrysophytes were all from freshwater, I would assume that P availability is the main driver (see Raven 2013, *J. of Exper. Bot.* 64:4023-4046)

P4 line 6. Trophic strategies (not trophy)

Line 12. Present study (not presented study)

P5. Line 11-14. The genomes were nowhere close to complete.

P5. Line 18. I am not convinced that the ploidy status and GC content would be independent of genome completeness. It seems that sequencing biases and repetitive regions could bias the results.

Line 20. These are rather high predicted gene counts, for example other Stramenopile flagellates range from 10K to 20K using haploid gene models. (Gobbler et al 2011, Wang et al 2014 and Radokovits et al 2012).

P7. Ploidy, more detail should be given in what is meant by ploidy and how it was estimated, the methods on this were rather brief (use of smudgpot, p.17, line 2-4). I have not seen this used for unicellular algae elsewhere.

P8. Line 6. From my search through the literature the *S. vulgaris* strain, was matched with an old description and a new type from the Antarctic was provided. There is a difference between the Arctic and the Antarctic.

P9. Line 1-6. There is no statistical support for anything in this paragraph of the discussion. Or elsewhere.

Line 15. why are chrysophytes assumed to be haploid? Most life cycle studies suggest that the freeliving stages and asexual stratospores are haploid at least in planktonic chrysophytes (see Sandgren, 1991 and others). If there is more recent literature on this please cite.

Reviewer #2 (Remarks to the Author):

Majda et al. sequenced 13 new chrysophyte genomes with illumina and/or PacBio platforms. Together with the three published chrysophyte genomes, they compared all the sequenced genomes in terms of gene sets, size, ploidy, coding density, and GC content, in order to see whether those characteristics are correlated with nutritional modes such as obligate phototroph, mixotroph, and obligate heterotroph. Through the comparative genomics, they conclude that nutritional constraints drive genome evolution. Comparative genomics of chrysophytes with different nutritional modes is a very outstanding strategy to investigate genome evolution driven by nutritional modes.

However, it is difficult to evaluate whether or not the comparison has been performed precisely given the low BUSCO values to evaluate the completeness and quality of assembled genomes. In addition, there is inconsistency between certain sentences in Results sections and those in Discussion section, making the manuscript confusing. Thus, I would say the conclusion is not strongly supported. Some findings in this paper are not novel but only support the previous studies. For example, three genomes of heterotrophic chrysophytes have been already published and analyzed. Using three heterotrophic chrysophyte genomes, ploidy analyses and conclusion that ploidy is independent from phylogeny have been already reported in the previous paper of the authors (Majda et al. 2019 GBE).

I respectfully recommend additional experiments and analyses to support the conclusion and discussion. I also recommend checking consistency of the authors' claims throughout the manuscript. Specific comments to be addressed are described below.

Major comments

1. BUSCO

Given the manuscript, the BUSCO analyses were performed directly with the genome sequences. I would recommend applying BUSCO to the predicted gene models, which would evaluate more precisely the completeness on the basis of presence or absence of genes conserved in protists, "protist dataset" the authors call. I would also recommend using the "eukaryotic conserved gene set" of BUSCO for evaluation not using the protist dataset. This is because the taxa to construct the protist dataset are restricted to particular lineages but not globally sampled. This might mislead to either over- or underestimate the quality of assembled genomes. The other reason is that previous studies of chrysophytes (Dorrell et al. 2019 PNAS; Majda et al. 2019 GBE) has evaluated the quality of transcriptome data and genomes, respectively, with the "eukaryotic conserved gene set." As the BUSCO scores for the transcriptome data ranged from 60 – 90% and those for the genomes were around 80%, they would be a criterion to evaluate the completeness of the genome data of

chrysophytes presented in this study.

2. Sequencing strategies

To make genome data and gene models more reliable, it would be better to reanalyze raw data.

A. Filtering

As mentioned above for BUSCO, I am concerned about the quality of genome assembly and the gene model. There is no information regarding quality filtering of raw reads before assembling. Read quality would strongly affect ploidy, assembly, prediction of gene models. It is better to clarify.

B. MetaBAT vs. MaxBin2

It is not clear to me why MaxBin2 has been chosen. Is it apparent that MetaBAT has artificially removed "much more reads"? Is it unlikely that MaxBin2 overestimates reads to keep but MetaBAT correctly removes more reads?

C. Assembly errors

Even after PacBio-based assembly, it is important to check whether there are errors left on nucleotide sequences, by using filtered illumina short reads. There is no information regarding inevitable steps for checking of those errors, such as Pilon. Please clarify.

D. Classification of contigs

The criteria 1 and 2 in P16 lines 4-7 does not make sense for me.

Please explain more in detail and clarify why these criteria can rationalize the classification.

E. Pathway analysis

In eukaryotes, different from prokaryotes, intracellular structures are complex and each compartment is separated by membranes. When reconstructing pathways, it should be considered whether those reactions are located in a same compartment. There is no information regarding targeting sequences at the N-termini for protein sequences involved in the pathway depicted in figure 3 and supplementary figures.

F. Phylogenetic analyses

Please clarify the model used. Maybe the GTR + G + I model? The analysis described was performed with the ultrafast bootstrap analysis. Please check bootstrap values with the canonical non-parametric bootstrap analysis.

3. biased sampling

The authors use three strains of the same species *P. lacustris*. I am afraid if those biased sampling might affect any results for comparative genomics among different nutritional modes. I would recommend removing two of three *P. lacustris* strains and check whether the removal affects results and conclusions.

4. Organellar genome evolution

Nutritional modes might most affect to plastid genomes as well as mitochondrial genomes. It would be interesting to perform the same analyses with those organellar genomes to see any relationship between nutritional modes and genome evolution. Although genome size and gene density in the

nuclear genomes are not so strongly correlated to changes of nutritional modes, organellar genomes might be more sensitive to that kind of changes. This is not so much labor as the assembled data would have already contain those small genomes.

5. GC content in P8

Overall GC% in coding regions are highly affected by species-specific codon frequency. I recommend to investigate GC% of 3rd codon positions for coding regions, which might provide a much clearer trend of GC% preference as seen in non-coding regions.

6. Inconsistency between Results and Discussion

A. P8 lines 18-21

The authors conclude in Discussion that they find genome reduction accompanied by an increase in the genome GC% and an increase in gene density due to evolutionary shift to heterotrophy from obligate phototrophy. However, some of those “findings” are not supported by any data in Results. Indeed, the authors describe in the Results section that “we refrain from drawing a clear conclusion” in Genome size (P7 line 6) and “differences were not significant” in Gene density (P7 line 26). It would be better to reconstruct the manuscript.

B. P11 line 27 – P12 line 1

“This is supported by an increased fraction of genes related to environmental information processing and secondary metabolism in the photo- and mixotrophic group (see Fig. 2) while the heterotrophic species have a more specialized and unique gene inventory in these functional groups.”

I am not sure which data in Fig. 2 support this idea. The gene assignment (%) for Secondary metabolism is not explicitly decreased in heterotrophs but highly conserved throughout all the strains and all the nutritional modes in Fig. 2. Speaking about Environmental information processing, actually three strains of *P. lacustris*, heterotrophic chrysophytes, possess smaller gene assignment (%) in Fig. 2A. As I point in Major comment 3, I am afraid if three strains of *P. lacustris* might have biasedly attracted “mean characteristics” of heterotrophic chrysophyte strains to the characteristics of *P. lacustris*, in this kind of comparison.

Rather, two strains of mixotrophs, *P. malhamensis* and *Ochromonas danica*, also show similar tendency to *P. lacustris* (Fig. 2A). I find those heterotrophs and mixotrophs are of clade 3 in Fig. 1, strongly suggesting lineage-specific characteristics in genomes but irrelevant to nutritional modes.

C. P9 lines 17 – 22

“Taking the different ploidies into account the genome size estimates based on flowcytometry and based on assembled genomes largely correspond except for few strains for which the ploidy level could not unambiguously determined (Table 3)”

“the genomes of phototrophic strains were considerably larger as compared to those of heterotrophic and mixotrophic strains”

It is not clear to me which data in Table 3 support the former sentence. Table 3 shows both total genome sizes and ploidies in addition to haploid genome sizes. Speaking about JBC27, it seems reasonable that the total genome size estimated by flowcytometry is 157 Mb as the haploid size estimated by genome sequencing is 82 Mb given the diploid genome estimated by k-mer frequency. Similar consistency can be seen in AR4D6.

However, for many others, total genome sizes estimated by the flowcytometry are not corresponding to calculated total genome sizes with the haploid sizes and ploidies. For example,

199hm has a 90 Mb haploid genome. As it is diploid as estimated by k-mer frequency, calculated total genome size would be around 180 Mb. However, the total genome size estimated by the flowcytometry is 293 Mb, much larger than 180 Mb estimated by the ploidy. It is better to clarify how the authors rationalize the inconsistency of the total genome sizes estimated by flowcytometry and by ploidy.

If one might trust the calculated total genome sizes with the haploid sizes and ploidies more than those estimated by the flowcytometry, total genome sizes of photosynthetic strains are not much larger than those of heterotrophic strains. Rather, total genome sizes of obligate heterotrophs are much larger than those of obligate phototrophs. In this case, the following sentence does not stand. P9 line 22

“the genomes of phototrophic strains were considerably larger as compared to those of heterotrophic and mixotrophic strains”

Minor points

1. ABSTRACT

“Phototrophic eukaryotes have evolved mainly by the primary or secondary uptake of photosynthetic organisms.”

I would say this sentence should be rephrased as it is controversial whether this is true. Some papers have proposed that red alga-derived plastids might be of tertiary or higher endosymbioses, except for those in cryptophytes which possess nucleomorphs, the evidence of secondary endosymbiosis (see e.g., Stiller et al. 2014 Nature Commun; Burki et al. 2016 Proc Soc Biol; Cenci et al. 2018 BMC Biol).

2. Phototrophy, mixotrophy, and heterotrophy

I think “phototrophy” is to be called “obligate phototrophy,” as mixotrophic organisms are tentatively phototrophic. Similarly, heterotrophy is to be called “obligate heterotrophy.”

Concerning nutritional modes, are all the heterotrophic chrysophytes phagotrophic? If so, it is interesting how the authors have accomplished to prepare “axenic” cultures without any food bacteria and how they confirm the culture are axenic. I imagine if a special medium has been used for the axenic cultures. Please clarify it in the Method section and Table 1.

3. Facultative pathways

What does “facultative pathways” exactly mean here?

4. Gold algae in keywords

Should be “golden algae.”

INTRODUCTION

5. P3 line1 and citation therein.

I am afraid I am wrong. But I could not find any description of “significance of nutritional constraints in the evolution of life” in the cited papers 1-3.

6. P3 line27 and citation therein

I could not find any description of plastid genomes and plastid-encoded genes of chrysophytes in Graupner et al.

7. P4 lines16-25

Although I am afraid if I am wrong, Hypothesis 1 might not stand in the first place.

Hypothesis 1 says the nuclear genome of obligate heterotrophic species is reduced in size than those of obligate phototrophic and mixotrophic species, as a result of nutrient limitations or as a size adaptation to their small food bacteria. In P3 line16, the authors define the nutrient limitation as limitation of nitrogen and phosphorus as well as other essential nutrients. For heterotrophs, the authors mention that prey or carbon shortage would be limitation instead of the nutrient limitation. As nucleotides contain more carbon molecules than phosphorus and nitrogen, carbon seems to be a stricter constraint on genome evolution. Otherwise, it would be difficult to define a cause of genome reduction only on the basis of limitation of nutrients or carbon. It is better to cite some pioneering papers for supporting the hypothesis in Introduction.

RESULTS

8. Table numbers should be in order. The current Table 1 should be "Table 5" when considering the order of appearance of tables. Similarly, the current Fig. 1 should be Fig. 4 as Fig. 2 and Fig. 3 appear prior to the current Fig. 1.

9. Genome size in P7

"Considering only genomes with high completeness"

I am afraid if the completeness might be inconclusive, given the low BUSCO scores such as less than 61%. It would be better to reevaluate the completeness as I recommend above.

10. Gene density in P7 line 23

"We expect to find a correlation between nutritional mode and gene density, especially in heterotrophs because of the strong selection towards smaller cells."

It is better to cite some papers to clarify the reason why the authors expect so.

Discussion

11. P8 line 26

It is not clear to me which data support the followings: "gene losses were predominately observed for non-annotated genes" and "while genes and pathways of the primary metabolism were kept in all lineages."

12. P9 line11

His \diamond has?

13. P10 line 4

"In protist, which replicate predominantly asexual"

Is it true? Any citation is needed.

14. P11 line 4

"polyploidy may also evolve under nutrient limitation"

I am not sure what this exactly means. Again, nutrient limitation is defined by the authors as limitation of phosphorus, nitrogen, and others. Polyploidy needs more P, N, and C as well as proteins binding DNA, and thus it seems unlikely that occurrence of polyploidy is relevant to nutrient limitations. Please clarify.

15. P11 line 22

What does “genes encoded in the plastid” exactly mean here?

I think Graupner et al. does not investigate plastid genomes.

16. P12 lines 4-8

“primary pathways and annotated pathways of the secondary metabolism are almost complete in the genome.”

“Gene reductions concern mostly genes which could not be annotated”

Data representing the findings should be indicated here. Which data are the evidence?

17. P12 line 11-13

“This completeness of pathways seems to be different for plastid-encoded genes as suggested by transcriptome data and supported by the structural reduction as observed in microscopical analyses.”

It is not clear to me what this exactly means. Does the “plastid-encoded genes” mean genes in a plastid genome? If so, transcriptome data do not suggest anything about completeness of genes in a plastid genome. It is not clear as well to me what the structural reduction supports. Please clarify.

18. P12 lines 15-17

“advantageous for breaking-up nutritional requirements in the case of food shortage”

“genes are still functional”

These could only be confirmed by further analyses after biochemical or genetic experiments.

Are those genes active in transcription during food shortage? It is better to show some data to support these.

19. P12 line 22

“Increased heterotrophy”

What does this exactly mean here? Please clarify.

20. “due to the small sample size”

This sounds a bit subjective. How can the authors exclude another possibility that there is no trend in relationship between gene density and nutritional modes?

21. P13 line1

“our estimates still seem reliable.”

It is unclear to me what the “our estimates” exactly means here and what makes “our estimates” reliable here. Please explain in more detail.

Figures

22. Figure 3

What does “genome + transcriptome” exactly mean here?

I cannot distinguish the colors of “transcriptome” and “genome + transcriptome.”

It is also difficult to follow which strains possess or lack a protein and which strains possess a protein in “transcriptome data” or in “genome + transcriptome data”. Please reconstruct the figure.

23. Table 2

of contigs of JBC07, JBM10, JBNZ41 should be 9.1, 9.4, and 13.8.

Reviewer #3 (Remarks to the Author):

This study presents the results of 16 strains of chryomonad flagellates whose genomes have been sequenced with the goal of testing the hypothesis that evolution in this protistan group, which encompasses species that are phototrophic, heterotrophic or mixotrophic (mixed phototrophic and heterotrophic nutrition), has been driven by differential availability/limitation of carbon and nutrients. While not a novel hypothesis, the data brought to bear to test this hypothesis are impressive. The analysis appears to be thorough, convincing, even-handed and insightful. I am highly supportive of publication.

I am in agreement with the basic interpretation of the results, and I have few substantive comments or criticisms to offer. They describe multiple interesting findings, which include: (1) Documentation of the loss of genes from the genome as chryomonads have switched (in an evolutionary sense) from phototrophy or mixotrophy to heterotrophy; (2) The unexpected finding that ploidy varied among the strains examined (from diploidy to tetraploidy), yet the ploidy of specific strains appears to be independent of phylogeny or nutrition; (3) Genome reduction in heterotrophic chryomonads has been carried out through the loss of different genes from different strains, a finding that the authors interpret as niche specialization among heterotrophic strains.

The trend towards low GC content in phototrophs and mixotrophs relative to heterotrophs is also an interesting finding, and one that has been observed in prokaryotes. As described in this study, it has been hypothesized to be related to the production of proteins with low nitrogen content but high carbon content relative to high GC genomes (e.g. Bragg & Hyder (2004), *Proc. Biol. Sci.* 271(Suppl. 5), S374–S377; Mende et al. (2017), *Nature Microbiol* 2:1367-1373). The interpretation of the correlation between GC content and carbon/nutrient limitation among chryomonads presented in this study is an interesting and potentially significant similarity.

Minor comments:

The comment on P.8, line 6 does not appear to make sense: “The strains *P. encystans* 1006 and *S. vulgaris* 199hm inhabit arctic regions. Both strains had the highest GC content (see Table 2)”. In fact, strain 199hm in Table 2 has the lowest GC content of all the strains, so either the value or the statement seems incorrect.

Also, P.8, line 14: The statement reads “nutritional mode had no significant impact” referring to effects on GC content. The initial paragraph states that GC content in heterotrophs was clearly higher than for mixotrophs and phototrophs. I think this statement must be referring to something

else, and needs sorting out, or perhaps I am misinterpreting. Some clarity is needed there.

Same page, line 15: “Besides, we did neither obtain correlations between assembly size nor the total genome size and the GC content”. This is an awkward sentence and might be rephrased as “Significant correlations were also not apparent between assembly size, or total genome size, and the GC content”.

P.9, line 5: “definitely” should probably be “definitively”.

P.9 line 7: Perhaps “are consistent with...” rather than “suggest it”.

P.9, line 10: The statement “Further, the genome size has been shown to correlate with cell volume (e.g., [15]), which his reduced in phagotrophic species preying on ultramicrobacteria.” Probably ‘his’ should be ‘is’, but the reference is a plant genome size paper, so it is not clear how that relates to predation on ultrabacteria. The latter part of the sentence appears to relate to the following sentence.

I suggest that Table 2 (and others) be expanded to include both the species names and the strain designations. Most readers will not have familiarity with the strain numbers. So, for example, in Tables 2, 3 and 4 it will be necessary for the reader to refer back to Table 1 for species names.

It may be more accurate to refer to ‘trophic mode’ rather than ‘nutrient mode’. Trophic mode implies a behavior (heterotrophy, mixotrophy, phototrophy) whereas nutrient mode is a bit vague. Mixotrophs may conduct prey consumption to obtain nutrients or carbon, so that could be confusing.

Some minor editing of the text is required.

We thank the reviewers for their time and constructive advices. The main changes in the new version of the manuscript are the improved BUSCO results based on more specific and recent datasets confirming the completeness of the genomes. Detailed answers to the reviewers including changes in the manuscript are given below:

Reviewer #1:

"A problem is relatively poor taxon sampling, with a heavy bias towards heterotrophic strains. The phototrophic strains both belong to more distantly related taxa and it seems that phylogeny might better explain the differences between the phototrophic and mixo-hetero species."

→ Actually, the species selection reflects the diversity of the chrysophytes. Most species are mixo- or heterotrophic and the few obligate phototrophic taxa branch early. Therefore only few phototrophic cultures exist, these were included in the genome analyses. The differentiation into mixo- and heterotrophic strains occurred several times independently and taxon sampling reflects these different independent differentiations into mixo- and heterotrophic strains.

"The sequenced phototrophic species were both scaly flagellates (Silica scales) and one could hypothesize that the use of silica covering results in loss of phagotrophy in general. The main evolutionary driver could be anti-predation (or something else) and nothing to do with nutrient limitation."

→ We disagree for two reasons: first, the scaly phototrophs branch early and scales may have been a plesiomorphy. Second and more importantly the scaled Paraphysomonadida are also heterotroph which shows that scales do not interfere with feeding. Of course we cannot exclude all possible characteristics, but we did mention in the current discussion that also other factors (like scales) could play a role in the differences between different nutritional modes.

Specific comments:

Introduction.

P3. Line 12. Even if a point modification was "random", for the resulting phenotype or trait to be maintained, there would have to be some evolutionary selection.

→ Here our wording was misleading, that's why we changed the sentence to *„In contrast to former individual case studies, here we use the parallel evolution of heterotrophic Chrysophyceae from phototrophic and mixotrophic ancestors in order to separate general directions and constraints in genome evolution from those related to the nutritional mode.“*

Line 15-19. Some linking (context) text seems to be missing. Explain how nutrient limitation could be related to "size"?

→ Thanks for the remark. We explain the relation between nutrient limitation and size in the introduction in more detail now. In brief, phototrophs need a minimum cell size for hosting a plastid (and further larger cells may benefit from their larger cell sizes in order to maximize light harvesting), whereas heterotrophs profit from smaller cells, which enhances effectiveness of predation and enables feeding from ultramicrobacteria due to better/optimal predator-prey size ratios.

Line 22. This would be a good place to explain what is meant by "costs" I assume this is relative content of N or P in nucleic acids or amino acids (codon usage). Since the sequenced

chrysophytes were all from freshwater, I would assume that P availability is the main driver (see Raven 2013, J. of Exper. Bot. 64:4023-4046)

- The current introduction explains the costs in more detail. It is correct that the P availability is one important factor. Further, cytosine contains one N more than thymine (GC more „expensive“ than AT). However, coding regions containing a higher GC result in „cheaper“ amino acids which may counterbalance the increased needs in the genome.

P4 line 6. Trophic strategies (not trophy)

Line 12. Present study (not presented study)

- Thank you for the remark, we have corrected the misspelling.

P5. Line 11-14. The genomes were nowhere close to complete.

- BUSCO is very strict. From our previous study (Majda 2019) we noticed that using the protists data set a 61% recovery rate indicates an almost complete chrysophyte genome. We repeated the BUSCO analysis with the Stramenopiles gene set, which gives an average completeness of 59%.

P5. Line 18. I am not convinced that the ploidy status and GC content would be independent of genome completeness. It seems that sequencing biases and repetitive regions could bias the results.

- . Ploidy is estimated on k-mer pairs (sequences of a certain length that differ by exactly one nucleotide) and repetitive regions consist of concatenations of very similar sequences therefore, in repetitive regions there are only few k-mer variations with high frequencies. The ploidy estimation usually checks over one million k-mer pairs, which is why repetitive regions in k-mer based methods are neglectable.

Even in incomplete genome assemblies the proportion of k-mer pairs is determined by allele variation or respectively ploidy. Sequencing errors are random events and can be compensated by a high coverage, which we predominantly had. However, as mentioned in the manuscript, we had two cases (*Pedospumella encystans 1006*, *Mallomonas annulata*) with weak results. Here, there was either a too low coverage or a too high amount of repetitiveness. The authors of the tool smudgeplot (Ranallo-Benavidez, Nat Commun, 2020) have stated the following: „ The coverage of the dataset must be sufficient for these methods to resolve the error peak with the haploid peak. In general, having at least 15x coverage per homolog for GenomeScope and 25x coverage per homolog for Smudgeplot is required. Smudgeplot works well under moderate heterozygosity and repetitiveness where the signal from heterozygous k-mer pairs is stronger than the signal from repetitive k-mer pairs.... Species with extreme heterozygosity and high repetitiveness, such as cotton and wheat, can confuse a Smudgeplot analysis. “In our study, 14 of the 16 strains have a coverage above 25, with the lowest coverage values being 14 and 20. The ploidy estimates are therefore supposed to be reliable..

For the GC content even a partial genome should provide a comparable value, since it gives an average value over the whole assembly. In all trophic groups we have performed additional PacBio sequencing to resolve repetitive regions by longer sequence reads, therefore we do not expect to have a bias in GC content due to possibly unresolved repetitive regions and further do not find differences in GC content between assemblies with or without PacBio sequencing.

Line 20. These are rather high predicted gene counts, for example other Stramenopile flagellates range from 10K to 20K using haploid gene models. (Gobbler et al 2011, Wang et al 2014 and Radokovits et al 2012).

→ We agree with the reviewer that the number of predicted genes is rather high. The gene prediction is based on the tool AUGUSTUS, which is very reliable. Prior to gene prediction repeat masking was used, since repeats can disturb gene prediction and increase the number of genes which might still be the case. Therefore, we clustered the genes to orthologous groups.

In related organisms the calculation of gene amount is based on a much smaller draft genome (Radokovits et al 2012) or proteomics (Gobler et al 2011), which can differ due to other biases (e.g. non-expressed genes, preference of highly-expressed genes). But we also admit that not all predicted genes might be functional. We added a paragraph on this in the discussion.

P7. Ploidy, more detail should be given in what is meant by ploidy and how it was estimated, the methods on this were rather brief (use of smudgpot, p.17, line 2-4). I have not seen this used for unicellular algae elsewhere.

→ We added more details to the method of ploidy estimation.

P8. Line 6. From my search through the literature the *S. vulgaris* strain, was matched with an old description and a new type from the Antarctic was provided. There is a difference between the Arctic and the Antarctic.

→ We agree with the reviewer. The strains *P. encystans* 1006 and *S. vulgaris* 199hm inhabit both antarctic regions. We have removed the separate consideration of the antarctic strains as they do not provide any additional insight, see comment from Reviewer 3.

P9. Line 1-6. There is no statistical support for anything in this paragraph of the discussion. Or elsewhere.

→ We observed significant differences in assembly size between all three trophic groups (anova p-value < 0.05). Dependence of trophic and genome size was shown in Olefeld et al. 2018. Variation of ploidy was found in different lineages as well as within one species, whereby independence of phylogeny seems likely. We are unfortunately not aware of any statistical test that can prove this independence.

Line 15. why are chrysophytes assumed to be haploid?

→ Most life cycle studies suggest that the free-living stages and asexual stratospores are haploid. This assumption goes back to life cycle studies of the 1970s and 80s. Even though these studies were restricted to few taxa and molecular evidence is largely missing they suggested that the free-living stages and asexual stratospores are haploid at least in planktonic chrysophytes (see Sandgren, 1991 and others). This is what we refer to – the recent molecular evidence, including this study does not generally support this view. Additionally, the assumption of haploidy was based on the lower cell and genome size of heterotrophs (Olefeld, 2018) and the associated advantage of preying on ultramicrobacteria.

Reviewer #2:

Major comments

1. BUSCO

Given the manuscript, the BUSCO analyses were performed directly with the genome sequences. I would recommend applying BUSCO to the predicted gene models, which would evaluate more precisely the completeness on the basis of presence or absence of genes conserved in protists, “protist dataset” the authors call. I would also recommend using the “eukaryotic conserved gene set” of BUSCO for evaluation not using the protist dataset. This is because the taxa to construct the protist dataset are restricted to particular lineages but not globally sampled. This might mislead to either over- or underestimate the quality of assembled genomes. The other reason is that previous studies of chrysophytes (Dorrell et al. 2019 PNAS; Majda et al. 2019 GBE) has evaluated the quality of transcriptome data and genomes, respectively, with the “eukaryotic conserved gene set.” As the BUSCO scores for the transcriptome data ranged from 60 – 90% and those for the genomes were around 80%, they would be a criterion to evaluate the completeness of the genome data of chrysophytes presented in this study.

→ We agree. We repeated the BUSCO analysis based on the Stramenopile gene set with predicted genes, which gives an average completeness of 59%. We additionally used the three strains from Majda (2019, GBE) for comparability (average completeness of 71%), which were sequenced with high coverage with Illumina and in addition with PacBio.

2. Sequencing strategies

To make genome data and gene models more reliable, it would be better to reanalyze raw data.

→ There is always room for improvement, especially with novel genomes. However, we have used some of the best tools that are currently available for hybrid assemblies and binning and already tested several alternatives. We believe that without improvements in these tools, reanalyzing would result at most in a minimal enhancement.

A. Filtering

As mentioned above for BUSCO, I am concerned about the quality of genome assembly and the gene model. There is no information regarding quality filtering of raw reads before assembling. Read quality would strongly affect ploidy, assembly, prediction of gene models. It is better to clarify.

→ We added the following details. First quality filtering and adapter removal was done by the sequencing provider. The provider filter options were:

low quality threshold = 10, N rate threshold = 0.1, quality system = sanger.

Further, we checked the quality with FastQC and did not detect low quality reads or adapter remnants in the sequences (illumina reads quality > 20, pacbio quality >10).

B. MetaBAT vs. MaxBin2

It is not clear to me why MaxBin2 has been chosen. Is it apparent that MetaBAT has artificially removed “much more reads”? Is it unlikely that MaxBin2 overestimates reads to keep but MetaBAT correctly removes more reads?

→ That could be the case. Since we have added another selection step it seems to work better with an overestimation and a subsequent reduction than already start with a low read amount.

Both tools have good benchmarks (Sczyrba, *Nat Methods*, 2017) and it is hard to say which works better in that case. Due to the fact that MaxBin2 finds significantly more reads classified as eukaryotic, which were missing in MetaBAT bins, MaxBin2 seemed to be more appropriate.

C. Assembly errors

Even after PacBio-based assembly, it is important to check whether there are errors left on nucleotide sequences, by using filtered illumina short reads. There is no information regarding inevitable steps for checking of those errors, such as Pilon. Please clarify.

→ SPAdes builds a graph from filtered Illumina reads and uses the PacBio-contigs for extension. Thereby, errors from PacBio are not considered, since the Illumina reads determine the contig sequence. In case of a PacBio-contig without an associated Illumina-built branch, there is no error correction. However, here likewise Pilon cannot improve the contig quality without Illumina read coverage.

D. Classification of contigs

The criteria 1 and 2 in P16 lines 4-7 does not make sense for me.

Please explain more in detail and clarify why these criteria can rationalize the classification.

→ The NCBI RefSeq database contains the following numbers of DNA sequences (26 march 2020):

- Fungi(371,703)
- Protists(98,754)
- Bacteria(1,175,000)
- Archaea(5,541)
- Viruses(6,065)

These are the organisms which might occur in our cultivation. By using cultivation in almost monocultures (filtering bacteria, selection for desired strain, suppression of other bacteria by feeding bacteria in case of heterotrophic strains) the risks of contamination were minimized. Fungi could be excluded with a high probability based on the contamination check by microscope. Additionally, the classification of reads as fungi, archaea or virus were extremely low. But, there are around 10 times more bacterial sequences than protist sequences. In case of a sample with a high amount of unclassified sequences it is more likely these reads are affiliated to not well studied organisms (protists, archaea, viruses). Archaea were unlikely growing to a sufficient amount under present culture conditions. Viruses would be possible, but due to their size they would only cause a very small contamination. Certainly, there is always low contamination and we have to set the accepted limit somewhere, which was done after we had reviewed the records manually. In this case, we chose that the number of unclassified contigs in a cluster must at least be twice the number of bacterial contigs or the number of eukaryotic contig at least half the number of the bacterial contigs to classify a cluster as protistan. The last criteria is based on the fact, that by chance an assignment to bacteria is more likely. This is also the reason why we lost a large part of plastid and mitochondria sequences. Considering the 10 times higher probability to classify a contig as a bacterium, the above mentioned factor two is still conservative. Additionally, we map the reads of the eukaryotic clusters (bins) back to contigs assigned as prokaryotic, resulting in a more stringent filtering. But most important, there are only few cases (less than ~5%) where the classification is not obvious and these filtering criteria apply.

E. Pathway analysis

In eukaryotes, different from prokaryotes, intracellular structures are complex and each compartment is separated by membranes. When reconstructing pathways, it should be considered whether those reactions are located in a same compartment. There is no information regarding targeting sequences at the N-termini for protein sequences involved in the pathway depicted in figure 3 and supplementary figures.

→ This is an interesting question from the physiological point of view. We are not aware that other genome reconstruction studies are pursuing this question. It is out of the scope of our study, but once our genome data is published, this would be an interesting aspect for future research.

We did not detect differences in the presence of mitochondrial and plastid targeting proteins between the trophic groups (data not shown). We further do not aim to „reconstruct“ the metabolic pathways but rather use KEGG pathways as an illustration of identified genes and potential functions. We clarified this in the new version of the manuscript.

F. Phylogenetic analyses

Please clarify the model used. Maybe the GTR + G + I model? The analysis described was performed with the ultrafast bootstrap analysis. Please check bootstrap values with the canonical non-parametric bootstrap analysis.

→ Our study does not aim for a recalculation of a new phylogeny, it is merely used for demonstrating the distribution of ploidy. The phylogeny agrees with Grossmann et al. (2016, J Eukaryot Microbiol) and Bock et al. (2017, Fottea), who used more elaborate methods.

3. biased sampling

The authors use three strains of the same species *P. lacustris*. I am afraid if those biased sampling might affect any results for comparative genomics among different nutritional modes. I would recommend removing two of three *P. lacustris* strains and check whether the removal affects results and conclusions.

→ Thanks for the remark. We repeated each statistical test with three strains of *P. lacustris* as well as including only one strain (JBM10). In all cases, this had only a minor effect on the result and showed no difference in significance values.

4. Organellar genome evolution

Nutritional modes might most affect to plastid genomes as well as mitochondrial genomes. It would be interesting to perform the same analyses with those organellar genomes to see any relationship between nutritional modes and genome evolution. Although genome size and gene density in the nuclear genomes are not so strongly correlated to changes of nutritional modes, organellar genomes might be more sensitive to that kind of changes. This is not so much labor as the assembled data would have already contain those small genomes.

→ We already looked at the mitochondrial and plastid genomes but discovered that unfortunately we lost several organelle genomes, or parts, during binning or probably DNA isolation since we selected large fragment sizes for PacBio sequencing and did not enrich for these organelles. The remaining organelle genomes are presumably be not sufficient for meaningful statistical tests.

5. GC content in P8

Overall GC% in coding regions are highly affected by species-specific codon frequency. I recommend to investigate GC% of 3rd codon positions for coding regions, which might provide a much clearer trend of GC% preference as seen in non-coding regions.

- We agree with the reviewer. The main findings (difference between the nutritional modes) had a clear significance with a p-value < 0.01 . We further investigated the third codon position, however, the statistical test between the trophic modes resulted in a p-value of 0.08 (Anova). Presumably the reduction of non-coding sequences are mainly responsible for evolutionary change in GC.

6. Inconsistency between Results and Discussion

A. P8 lines 18-21

The authors conclude in Discussion that they find genome reduction accompanied by an increase in the genome GC% and an increase in gene density due to evolutionary shift to heterotrophy from obligate phototrophy. However, some of those “findings” are not supported by any data in Results. Indeed, the authors describe in the Results section that “we refrain from drawing a clear conclusion” in Genome size (P7 line 6) and “differences were not significant” in Gene density (P7 line 26). It would be better to reconstruct the manuscript.

- We decided to discard the part on gene density in the discussion. We think there is a trend, but we were not able to show significant differences.

B. P11 line 27 – P12 line 1

“This is supported by an increased fraction of genes related to environmental information processing and secondary metabolism in the photo- and mixotrophic group (see Fig. 2) while the heterotrophic species have a more specialized and unique gene inventory in these functional groups.”

I am not sure which data in Fig. 2 support this idea. The gene assignment (%) for Secondary metabolism is not explicitly decreased in heterotrophs but highly conserved throughout all the strains and all the nutritional modes in Fig. 2. Speaking about Environmental information processing, actually three strains of *P. lacustris*, heterotrophic chrysophytes, possess smaller gene assignment (%) in Fig. 2A. As I point in Major comment 3, I am afraid if three strains of *P. lacustris* might have biasedly attracted “mean characteristics” of heterotrophic chrysophyte strains to the characteristics of *P. lacustris*, in this kind of comparison.

Rather, two strains of mixotrophs, *P. malhamensis* and *Ochromonas danica*, also show similar tendency to *P. lacustris* (Fig. 2A). I find those heterotrophs and mixotrophs are of clade 3 in Fig. 1, strongly suggesting lineage-specific characteristics in genomes but irrelevant to nutritional modes.

- We changed “(see Fig. 2)” to “(see Fig. 2B, Core genes)” and added additional explanations. There is an increased fraction of genes related to environmental information processing and secondary metabolism in the phototrophic group and respectively genes related to environmental information processing in the mixotrophic group. The total amount of orthologous groups of the heterotrophic core genes group is far smaller than the amount of the photo- and mixotrophic group. We added that the lineage-specific characteristic is strong for close relatives (clade 3), but not significant in a wider degree of relationship. The influence of axenic growth and PacBio-sequencing seem to be further critical factor. However, despite this outlier clade, we could demonstrate characteristics of the nutritional modes.

C. P9 lines 17 – 22

“Taking the different ploidy levels into account the genome size estimates based on flow cytometry and based on assembled genomes largely correspond except for few strains for which the ploidy level could not unambiguously be determined (Table 3)”

“the genomes of phototrophic strains were considerably larger as compared to those of heterotrophic and mixotrophic strains”

It is not clear to me which data in Table 3 support the former sentence. Table 3 shows both total genome sizes and ploidy levels in addition to haploid genome sizes. Speaking about JBC27, it seems reasonable that the total genome size estimated by flow cytometry is 157 Mb as the haploid size estimated by genome sequencing is 82 Mb given the diploid genome estimated by k-mer frequency. Similar consistency can be seen in AR4D6.

However, for many others, total genome sizes estimated by the flow cytometry are not corresponding to calculated total genome sizes with the haploid sizes and ploidy levels. For example, 199hm has a 90 Mb haploid genome. As it is diploid as estimated by k-mer frequency, calculated total genome size would be around 180 Mb. However, the total genome size estimated by the flow cytometry is 293 Mb, much larger than 180 Mb estimated by the ploidy. It is better to clarify how the authors rationalize the inconsistency of the total genome sizes estimated by flow cytometry and by ploidy.

If one might trust the calculated total genome sizes with the haploid sizes and ploidy levels more than those estimated by the flow cytometry, total genome sizes of photosynthetic strains are not much larger than those of heterotrophic strains. Rather, total genome sizes of obligate heterotrophs are much larger than those of obligate phototrophs. In this case, the following sentence does not stand.

P9 line 22

“the genomes of phototrophic strains were considerably larger as compared to those of heterotrophic and mixotrophic strains”

- Since some genomes could not be completely sequenced or repeat regions could not be completely resolved, we used the genome size from flow cytometry as a more reliable source. We have changed the script accordingly. Additionally, we corrected three conveying errors (the genomes of the three strains of *P. lacustris* were factor 2 too large).

Minor points

1. ABSTRACT

“Phototrophic eukaryotes have evolved mainly by the primary or secondary uptake of photosynthetic organisms.”

I would say this sentence should be rephrased as it is controversial whether this is true. Some papers have proposed that red alga-derived plastids might be of tertiary or higher endosymbioses, except for those in cryptophytes which possess nucleomorphs, the evidence of secondary endosymbiosis (see e.g., Stiller et al. 2014 Nature Commun; Burki et al. 2016 Proc Soc Biol; Cenci et al. 2018 BMC Biol).

- That's true, in case of some dinoflagellate taxa also tertiary or serial secondary uptake have taken place. Since we are referring to the predominant uptake, we see no contradiction here.

2. Phototrophy, mixotrophy, and heterotrophy

I think “phototrophy” is to be called “obligate phototrophy,” as mixotrophic organisms are tentatively phototrophic. Similarly, heterotrophy is to be called “obligate heterotrophy.”

Concerning nutritional modes, are all the heterotrophic chrysophytes phagotrophic? If so, it is

interesting how the authors have accomplished to prepare “axenic” cultures without any food bacteria and how they confirm the culture are axenic. I imagine if a special medium has been used for the axenic cultures. Please clarify it in the Method section and Table 1.

- We added following sentence in the introduction: „ In this paper we refer to obligate heterotrophy and phototrophy if not stated otherwise.“ All heterotrophic chrysophytes are phagotrophic. The obtaining of axenic cultures is very challenging, which is why we only succeeded for a few strains. We added information about the medium to the supplement.

3. Facultative pathways

What does “facultative pathways” exactly mean here?

- We defined facultative pathways as pathways, which are not essential since the metabolites could be taken up otherwise (phagocytosis).

4. Gold algae in keywords

Should be “golden algae.”

- We corrected the phrase.

INTRODUCTION

5. P3 line1 and citation therein.

I am afraid I am wrong. But I could not find any description of “significance of nutritional constraints in the evolution of life” in the cited papers 1-3.

- The listed references show an exemplary overview of the numerous losses of photosynthesis. It is significant because in all protist phyla (which had a plastid) development occurred from phototrophy towards heterotrophy.

6. P3 line27 and citation therein

I could not find any description of plastid genomes and plastid-encoded genes of chrysophytes in Graupner et al.

- The plastid genomes were not subject of the study. In Graupner et al. the entire transcriptome has been investigated, including plastid-encoded respectively plastid-targeting genes. Comparing strains of different nutritional modes Graupner demonstrated varying completeness of the plastid by pathway analyses.

7. P4 lines16-25

Although I am afraid if I am wrong, Hypothesis 1 might not stand in the first place.

Hypothesis 1 says the nuclear genome of obligate heterotrophic species is reduced in size than those of obligate phototrophic and mixotrophic species, as a result of nutrient limitations or as a size adaptation to their small food bacteria. In P3 line16, the authors define the nutrient limitation as limitation of nitrogen and phosphorus as well as other essential nutrients. For heterotrophs, the authors mention that prey or carbon shortage would be limitation instead of the nutrient limitation. As nucleotides contain more carbon molecules than phosphorus and nitrogen, carbon seems to be a stricter constraint on genome evolution. Otherwise, it would be difficult to define a cause of genome reduction only on the basis of limitation of nutrients or carbon. It is better to cite some pioneering papers for supporting the hypothesis in Introduction.

- Nucleotides contain more carbon than phosphorus and nitrogen, but carbon also has a higher availability. Hence, the ratio of C:N:P is the crucial factor (e.g. Raven 2013, J. of

Exper. Bot. 64:4023-4046). In aquatic environments phototrophic organisms are usually limited by inorganic nutrients such as phosphate as the concentration of inorganic carbon (mostly in the form of hydrogen carbonate) for photosynthesis is usually not limiting. In contrast, heterotrophs take up organic molecules which are used as substrates for respiration. Heterotrophs are therefore often limited by the availability of prey (and organic molecules for respiration) corresponding usually to a carbon limitation while nutrient uptake in form of amino acids or nucleotides is less / not limiting.

RESULTS

8. Table numbers should be in order. The current Table 1 should be "Table 5" when considering the order of appearance of tables. Similarly, the current Fig. 1 should be Fig. 4 as Fig. 2 and Fig. 3 appear prior to the current Fig. 1.

→ Thanks, we reordered the figures and tables accordingly.

9. Genome size in P7

"Considering only genomes with high completeness"

I am afraid if the completeness might be inconclusive, given the low BUSCO scores such as less than 61%. It would be better to reevaluate the completeness as I recommend above.

→ As mentioned above, we have repeated the analysis, which shows that completeness is adequate for novel genomes.

10. Gene density in P7 line 23

"We expect to find a correlation between nutritional mode and gene density, especially in heterotrophs because of the strong selection towards smaller cells."

It is better to cite some papers to clarify the reason why the authors expect so.

→ Predator-prey theory suggests an optimal volumetric size ratio of 1:1000 or less for interception feeding organisms (Hansen et al. 1994). In conclusion, nanoflagellates are much too large to successfully prey on ultramicrobacteria (Boenigk et al. 2004). It has been suggested that a reduction of cell size is beneficial with respect to the predator-prey interaction but may require the reduction of intracellular structures such as the plastid (de Castro et al. 2009) and possibly of the genome size (Olefeld et al. 2018). One possibility of size reduction would be the deletion of non-coding regions leading to higher gene density.

Discussion

11. P8 line 26

It is not clear to me which data support the followings: "gene losses were predominately observed for non-annotated genes" and "while genes and pathways of the primary metabolism were kept in all lineages."

→ In Fig. 2B (on the right side) the amount of orthologous groups (OGs) is separated in annotated and non-annotated genes. The pan genome shows little difference in annotated OGs, whereas non-annotated OGs differ stronger, for example between the mixotrophic group (M) and mixotrophic + heterotrophic group (M+H). In the core genome the amount of non-annotated OGs is in relation stronger reduced than the annotated OGs. The supplement figures 7 to 12 show several pathways of the primary metabolism.

12. P9 line11

His □ has?

→ We checked and removed this spelling mistake.

13. P10 line 4

“In protist, which replicate predominantly asexual”

Is it true? Any citation is needed.

→ On theoretical grounds based on the genetic advantages of recombination, it has been argued that most protists are nevertheless likely cryptically sexual at least occasionally (Dunthorn and Katz 2010; Hofstatter and Lahr 2019). And on experimental grounds evidence for this cryptic sex throughout the protists has been found by inventorying meiotic genes in different putative asexual lineages (Ramesh et al. 2005; Malik et al. 2008; Chi et al. 2014b; Dunthorn et al. 2017; Hofstatter et al. 2018; Kraus et al. 2018), although these meiotic genes could be used just for selfing or for non-canonical genetic pathways (Dunthorn et al. 2017). Sex has, however, be lost in some lineages, some of which could be ancient (Doerder 2014).

14. P11 line 4

“polyploidy may also evolve under nutrient limitation”

I am not sure what this exactly means. Again, nutrient limitation is defined by the authors as limitation of phosphorus, nitrogen, and others. Polyploidy needs more P, N, and C as well as proteins binding DNA, and thus it seems unlikely that occurrence of polyploidy is relevant to nutrient limitations. Please clarify.

→ Polyploidy would be affected from any limitation regarding the costs of genome maintenance. We think a nutrient limitation could be a multiplier for evolution. Despite the higher cost of P, N and C, there are advantages of polyploidy that might outweigh these energy costs. For example, the higher gene copy number leads to the possibility of retaining gene functions and simultaneously evolving genes (Predator-prey theory; Comai 2005). Ploidy also prevents the accumulation of mutations (Muller's ratchet; Muller 1932).

15. P11 line 22

What does “genes encoded in the plastid” exactly mean here?

I think Graupner et al. does not investigate plastid genomes.

→ The transcriptomes in the study of Graupner et al. also contained transcripts from the plastid.

16. P12 lines 4-8

“primary pathways and annotated pathways of the secondary metabolism are almost complete in the genome.”

“Gene reductions concern mostly genes which could not be annotated”

Data representing the findings should be indicated here. Which data are the evidence?

- Fig. 2 and table 5 show the proportion of annotated genes. The supplemental figures. S7-12 indicate the pathway completeness.

17. P12 line 11-13

“This completeness of pathways seems to be different for plastid-encoded genes as suggested by transcriptome data and supported by the structural reduction as observed in microscopical analyses.”

It is not clear to me what this exactly means. Does the “plastid-encoded genes” mean genes in a plastid genome? If so, transcriptome data do not suggest anything about completeness of genes in a plastid genome. It is not clear as well to me what the structural reduction supports. Please clarify.

- The phrase „plastid-encoded genes” means genes in the plastid genome. As mentioned above: The transcriptomes in the study of Graupner et al. also contained transcripts from the plastid. In our study we could unfortunately neither construct the plastid nor its pathways.

18. P12 lines 15-17

“advantageous for breaking-up nutritional requirements in the case of food shortage”

“genes are still functional”

These could only be confirmed by further analyses after biochemical or genetic experiments. Are those genes active in transcription during food shortage? It is better to show some data to support these.

- This advantage is hypothetical. We qualify our statement by indicating that complete pathways are not synonymous to functional genes and further tests are necessary. However, the completeness of the pathways makes the functionality more likely.

19. P12 line 22

“Increased heterotrophy”

What does this exactly mean here? Please clarify.

- We changed it to „advanced heterotrophy“.

20. “due to the small sample size”

This sounds a bit subjective. How can the authors exclude another possibility that there is no trend in relationship between gene density and nutritional modes?

- The reviewer is correct and possibly there is no trend in the relationship. But what we statistically test is a clear distinction between the trophic groups which might not be the case if we consider a transition towards heterotrophy and the different stages in reduction seen in the transcriptomic and microscopy data (Graupner et al. 2018, Grossman et al. 2016). This trend is visible in Fig. 4, but as stated not statistically different based on the distinction into 3 groups. More sequenced genomes could clarify this in future studies.

21. P13 line1

“our estimates still seem reliable.”

It is unclear to me what the “our estimates” exactly means here and what makes “our estimates” reliable here. Please explain in more detail.

- The estimates refer to the gene density of the investigated chrysophytes. The gene prediction is based on the tool AUGUSTUS, which is very reliable. Gene densities of

related organisms are mostly based on different methods (transcriptomic data or proteomics data), which can differ from genomic gene density estimates due to unexpressed genes in the transcriptome or non-functional genes in the genome. Additionally, there are eukaryotes with gene densities similar to bacteria, which could also be possible for the investigated strains.

Figures

22. Figure 3

What does “genome + transcriptome” exactly mean here?

I cannot distinguish the colors of “transcriptome” and “genome + transcriptome.”

It is also difficult to follow which strains possess or lack a protein and which strains possess a protein in “transcriptome data” or in “genome + transcriptome data”. Please reconstruct the figure.

- The phrase “genome + transcriptome” means, that the genes were found in both, the genomic and transcriptomic data sets. We edited the caption accordingly. Since the detection of a gene in the transcriptome automatically includes the genome, groups “genome + transcriptome” and “transcriptome” have equivalent informational content and were both indicated in blue. In contrast, newly found genes that were only found in the genome are marked in red.

23. Table 2

of contigs of JBC07, JBM10, JBNZ41 should be 9.1, 9.4, and 13.8.

The numbers were changed accordingly.

Reviewer #3 (Remarks to the Author):

This study presents the results of 16 strains of chryomonad flagellates whose genomes have been sequenced with the goal of testing the hypothesis that evolution in this protistan group, which encompasses species that are phototrophic, heterotrophic or mixotrophic (mixed phototrophic and heterotrophic nutrition), has been driven by differential availability/limitation of carbon and nutrients. While not a novel hypothesis, the data brought to bear to test this hypothesis are impressive. The analysis appears to be thorough, convincing, even-handed and insightful. I am highly supportive of publication.

I am in agreement with the basic interpretation of the results, and I have few substantive comments or criticisms to offer. They describe multiple interesting findings, which include: (1) Documentation of the loss of genes from the genome as chryomonads have switched (in an evolutionary sense) from phototrophy or mixotrophy to heterotrophy; (2) The unexpected finding that ploidy varied among the strains examined (from diploidy to tetraploidy), yet the ploidy of specific strains appears to be independent of phylogeny or nutrition; (3) Genome reduction in heterotrophic chryomonads has been carried out through the loss of different genes from different strains, a finding that the authors interpret as niche specialization among heterotrophic strains.

The trend towards low GC content in phototrophs and mixotrophs relative to heterotrophs is also an interesting finding, and one that has been observed in prokaryotes. As described in this study, it has been hypothesized to be related to the production of proteins with low nitrogen content but high carbon content relative to high GC genomes (e.g. Bragg & Hyder (2004), *Proc. Biol. Sci.* 271(Suppl. 5), S374–S377; Mende et al. (2017), *Nature Microbiol* 2:1367-1373). The interpretation

of the correlation between GC content and carbon/nutrient limitation among chrysoomonads presented in this study is an interesting and potentially significant similarity.

Minor comments:

The comment on P.8, line 6 does not appear to make sense: "The strains *P. encystans* 1006 and *S. vulgaris* 199hm inhabit arctic regions. Both strains had the highest GC content (see Table 2)". In fact, strain 199hm in Table 2 has the lowest GC content of all the strains, so either the value or the statement seems incorrect.

- We corrected the mistake. The strains *P. encystans* 1006 showed the highest and *S. vulgaris* 199hm showed the lowest GC value among the heterotrophic strains. We dropped the sentence, since they weigh each other out in statistics.

Also, P.8, line 14: The statement reads "nutritional mode had no significant impact" referring to effects on GC content. The initial paragraph states that GC content in heterotrophs was clearly higher than for mixotrophs and phototrophs. I think this statement must be referring to something else, and needs sorting out, or perhaps I am misinterpreting. Some clarity is needed there.

- The GC content regarding coding regions, non-coding regions or introns was independent of the nutritional mode. We rephrased the sentence.

Same page, line 15: "Besides, we did neither obtain correlations between assembly size nor the total genome size and the GC content". This is an awkward sentence and might be rephrased as "Significant correlations were also not apparent between assembly size, or total genome size, and the GC content".

- We rephrased the sentence as proposed.

P.9, line 5: "definitely" should probably be "definitively".

- We corrected the misspelling.

P.9 line 7: Perhaps "are consistent with..." rather than "suggest it".

- Thanks for the advice, but we prefer "suggest" as it better indicates that we are uncertain here.

P.9, line 10: The statement "Further, the genome size has been shown to correlate with cell volume (e.g., [15]), which his reduced in phagotrophic species preying on ultramicrobacteria." Probably 'his' should be 'is', but the reference is a plant genome size paper, so it is not clear how that relates to predation on ultrabacteria. The latter part of the sentence appears to relate to the following sentence.

- We corrected the misspelling. In plants the genome size correlates with cell size.

I suggest that Table 2 (and others) be expanded to include both the species names and the strain designations. Most readers will not have familiarity with the strain numbers. So, for example, in Tables 2, 3 and 4 it will be necessary for the reader to refer back to Table 1 for species names.

→ Thanks for the advice, we added the species names to the tables.

It may be more accurate to refer to 'trophic mode' rather than 'nutrient mode'. Trophic mode implies a behavior (heterotrophy, mixotrophy, phototrophy) whereas nutrient mode is a bit vague. Mixotrophs may conduct prey consumption to obtain nutrients or carbon, so that could be confusing.

→ Thanks for the hint, we will preferably use the expression 'trophic mode' .

Some minor editing of the text is required.

REVIEWERS' COMMENTS:

Reviewer #1 (Remarks to the Author):

The authors have addressed most of the comments by the reviewers. There are still a number of small grammar -typographical errors that can be easily addressed.

I would request that the authors define what they mean by ultramicrobacteria. The two references given at first mention do not define this. Please state the size. My experience is that the smallest mixotrophs can graze on quite large bacteria that occur in pelagic systems (large for me anyway, being 0.6 to 1.2 microns). The whole predator to prey size theory is misleading, keep in mind there are a lot of ways to capture prey, for example dinoflagellates can engulf whole chains of diatoms.

Or do they mean members of the Candidate Phyla Radiation (Hug, L. A., B. J. Baker, K. Anantharaman, and others. 2016. A new view of the tree of life. *Nat.*313 *Microbiol.* 1: 16048. doi:10.1038/nmicrobiol.2016.48). Few or none of which are currently in culture.

I would also disagree that having a chloroplast prevents cells from being "small". The smallest eukaryotes on record (except for maybe *Picomonas*, are Photosynthetic chlorophytes (*Ostreococcus* and *Bathycoccus*). In addition there are small mixotrophs in the Pavlovales and cryptophyceae. Maybe a sentence qualifying the remark or limiting it to ochrophytes would be advisable.

Reviewer #2 (Remarks to the Author):

All of my concerns raised for the previous manuscript have been well addressed, and I do not have any further comment except for correction of typos; some typos have not yet been corrected.